# Dopamine signaling regulates predator-driven changes in *Caenorhabditis elegans'* egg laying behavior

**Amy Pribadi[1,2†], Michael A Rieger[2†], Kaila Rosales[2], Kirthi C Reddy[2], Sreekanth H Chalasani[1,2]***

[1]Biological Sciences Graduate Program, University of California, San Diego, La Jolla, United States; [2]Molecular Neurobiology Laboratory, The Salk Institute for Biological Studies, La Jolla, United States

**Abstract** Prey respond to predators by altering their behavior to optimize their own fitness and survival. Specifically, prey are known to avoid predator-occupied territories to reduce their risk of harm or injury to themselves and their progeny. We probe the interactions between *Caenorhabditis elegans* and its naturally cohabiting predator *Pristionchus uniformis* to reveal the pathways driving changes in prey behavior. While *C. elegans* prefers to lay its eggs on a bacteria food lawn, the presence of a predator inside a lawn induces *C. elegans* to lay more eggs away from that lawn. We confirm that this change in egg laying is in response to bites from predators, rather than to predatory secretions. Moreover, predator-exposed prey continue to lay their eggs away from the dense lawn even after the predator is removed, indicating a form of learning. Next, we find that mutants in dopamine synthesis significantly reduce egg laying behavior off the lawn in both predator-free and predator-inhabited lawns, which we can rescue by transgenic complementation or supplementation with exogenous dopamine. Moreover, we find that dopamine is likely released from multiple dopaminergic neurons and requires combinations of both D1- (DOP-1) and D2-like (DOP-2 and DOP-3) dopamine receptors to alter predator-induced egg laying behavior, whereas other combinations modify baseline levels of egg laying behavior. Together, we show that dopamine signaling can alter both predator-free and predator-induced foraging strategies, suggesting a role for this pathway in defensive behaviors.

*For correspondence:
schalasani@salk.edu

†These authors contributed equally to this work

**Competing interest:** The authors declare that no competing interests exist.

## Editor's evaluation

In this important paper, Pribaldi and colleagues provide convincing evidence that locomotor and egg-laying behaviors of the nematode *C. elegans* can be altered by predation. They provide solid evidence that the neuromodulator dopamine is important for predator-evoked behavior, though further work will be necessary to understand how predator exposure might alter dopamine signaling. Because of the novelty of the behavioral findings and some important mechanistic insight, this work significantly advances the understanding of *C. elegans* neuroethology.

## Introduction

Predator-prey systems offer a rich variety of prey behaviors to explore, from innate to learned responses. Prey responses to predators also vary depending on the predation strategy (*Belgrad and Griffen, 2016*; *Palmer and Packer, 2021*), as well as the prey's abilities and the environmental context of both species (*Garcia and Koelling, 1966*). Additionally, prey can evaluate the cost/benefit of engaging in these antipredator behaviors, since they might impose additional costs by reducing access to food or

mates (*Kavaliers and Choleris, 2001*). While predators kill and consume prey, they can also influence prey behavior without necessarily inflicting direct harm, in both wild and laboratory contexts (*Lima, 1998*; *Sih, 1980*). However, these changes in prey behavior often involve costs like reduced access to food or mates (*Sih, 1980*). For example, reintroducing wolves into Yellowstone National Park resulted in changes to the grazing patterns of female elks with calves, with more time devoted to vigilance behaviors (*Childress and Lung, 2003*; *Laundré et al., 2001*). In the laboratory setting, rats presented with cat odor spent more time in shelter than exploring, feeding, or mating (*Choi and Kim, 2010*; *Kim et al., 2018*). Laboratory experiments in model organisms can lack the natural context of predator-prey dynamics, but observation in the wild lacks the ability to link predator-prey behaviors to molecules and neural pathways. To bridge the gap between ecological relevance and mechanistic insight, we explored a predator-prey system in nematodes that brings a naturalistic predator-prey interaction into the laboratory, making it more amenable to controlled experimentation.

*Caenorhabditis elegans* is a nematode that lives in rotting vegetation and eats the bacteria found there (*Schulenburg and Félix, 2017*). With 302 neurons and a mapped connectome (*White et al., 1986*), it is a model well suited to study behavior with the manipulation of genes and circuits often at the resolution of a single cell. While much research in predator-prey relationships involve organisms that have vision, little is known about defensive behaviors in olfactory/mechanosensory-dependent organisms like *C. elegans*. With different dependencies on sensory modalities, *C. elegans*-specific behaviors may not necessarily mimic defensive behavior traditionally associated with sighted prey, such as freezing (*Yilmaz and Meister, 2013*; *De Franceschi et al., 2016*). *C. elegans* spends most of its time searching for food or eating it, as well as laying eggs, so predator threat may influence these activities. The motor sequences required for changes to navigation when searching for food, such as the frequency of turns and reversals, are subject to the integration of input from several sensory neurons, and their modulation by biogenic amine neurotransmitter signaling (*Gray et al., 2005*; *Hills et al., 2004*). Although non-predative, there are numerous examples of *C. elegans* altering this system of navigational decision making in response to encounters with potentially aversive stimuli. For example, *C. elegans* will learn to avoid pathogenic bacteria such as *Serratia marcescens*, a behavior mediated by serotonin signaling (*Zhang et al., 2005*). *C. elegans* will also sense and navigate away from certain metal ions such as $Cu^{2+}$, and neurons mediating $Cu^{2+}$ response are modulated in turn by octopamine and serotonin (*Guo et al., 2015*). This response is also enhanced by the presence of food which is mediated by dopaminergic signaling (*Ezcurra et al., 2011*). Dopaminergic signaling also impacts how an animal locomotes in response to mechanically aversive stimuli such as the touch response, which itself is again modified by the presence of food (*Kindt et al., 2007*). These same neurotransmitters also impact the decision of when and where to lay eggs. Exogenous serotonin is known to promote the rate of egg laying off food, meanwhile exogenous octopamine and tyramine can inhibit this behavior (*Alkema et al., 2005*). Dopaminergic signaling couples locomotor behavior and egg laying, promoting the rate of egg laying when animals are roaming (*Cermak et al., 2020*). Like other potentially aversive stimuli, predator responses may be expected to modify how an animal navigates its environment. Like these stimuli, predator-evoked changes to exploration would likely intersect with the availability of food, potentially impacting activities like egg laying, all of which is expected to be modulated by biogenic amine neurotransmitter and receptor signaling pathways.

Previous studies have shown that nematodes of the *Pristionchus* genus can predate on other nematodes like *C. elegans* (*Sommer, 2006*) and are found in necromenic association with beetles (*Hong and Sommer, 2006*; *Herrmann et al., 2006*) as well as in rotting vegetation along with *Caenorhabditis* (*Félix et al., 2018*; *Félix et al., 2013*). Members of the *Pristionchus* genus exhibit mouth polyphenism, with either two-toothed 'eurystomatous' (Eu) or one-toothed 'stenostomatous' (St) mouthforms (*von Lieven and Sudhaus, 2000*; *Sudhaus et al., 2003*). The Eu mouthform in *P. pacificus* has been shown to enable more successful killing of nematode prey like *C. elegans* (*Serobyan et al., 2014*; *Wilecki et al., 2015*). While *P. pacificus* is a relatively well-studied species within *Pristionchus,* it is uncertain whether *C. elegans* actually interacts with *P. pacificus* in nature. In contrast, the gonochoristic species *Pristionchus uniformis* has been found in the same sample with wild *C. elegans* isolates (*Félix et al., 2018*), thus *P. uniformis* may represent a likelier candidate for naturalistic predative antagonism to *C. elegans*. Although *P. uniformis* was first characterized as a St-only species (*Fedorko and Stanuszek, 1971*), it has recently been re-assessed and found to possess both a bacterivorous St and the predatory

Eu mouthform (*Kanzaki et al., 2014*), and we too find that in standard growth conditions most *P. uniformis* strain JU1051 individuals have an Eu mouthform (*Figure 1—figure supplement 1*).

To test the hypothesis that, like other aversive stimuli, predators were able to exert an influence on patterns of *C. elegans* exploration, we wondered whether we could observe changes to *C. elegans* position and egg laying relative to food when animals experienced predator threat, and how factors like predator presence and bacterial topology intersect. As navigation and egg laying are influenced by biogenic amine signaling, we also wondered whether we could then use the molecular tools developed in *C. elegans* to discover the mechanisms underlying any observed changes to behavior. In this study, we show that *C. elegans* avoids a bacterial lawn that is occupied by its naturally cohabiting predator *P. uniformis* (*Félix et al., 2018*), and lays its eggs away from that lawn. We find that predator-exposed *C. elegans* potentiates the probability of egg laying off of the lawn, and this effect is sustained for many hours even after the predator is removed. This potentiation is further exaggerated when food is present outside the main bacterial lawn. Furthermore, we find that *C. elegans* egg laying locations are regulated by biogenic amine signaling in both baseline and predator-exposed conditions. Complete loss of dopamine synthesis resulted in overall reductions to egg laying at off lawn locations, which was restored by supplementation with exogenous dopamine. However, loss of signaling through combinations of D1 and D2 receptor homologs was able to perturb predator-induced off lawn egg laying behavior while maintaining baseline levels. Taken together we present a framework for interrogating prey behavior in nematodes, define some of the dynamics of this behavior, and identify potential molecular regulators of egg laying under predator threat.

## Results

### *C. elegans* avoids bacterial lawns inhabited by *Pristionchus* predators

We recently showed that *P. pacificus* bites *C. elegans* adults even though it is difficult to consume them. This biting of adult *C. elegans* prey forces these animals to leave the bacterial lawn, resulting in more exclusive access to the lawn by the predator (*Quach and Chalasani, 2022*). Using a modified version of the protocol in our previous study (*Quach and Chalasani, 2022*), we placed three predators and three *C. elegans* on an assay plate containing a small, dense bacterial lawn. Animals were restricted to an arena that included the lawn and a small area of empty agar (see Materials and methods). Control plates ('mock') had six *C. elegans* to maintain a consistent number of animals between plates with and without predators. These behavioral arenas were imaged under various experimental conditions, and coordinates of the eggs in arenas were determined. These coordinates were used to compute the distances of individual eggs from lawn center as well as their position relative to the lawn edge. Since *Pristionchus* also lay eggs, we used a *C. elegans* strain that expresses the GFP fluorophore in all of its eggs (*Pelt-2*::GFP) (*Figure 1a*).

To observe whether predator biting affects *C. elegans* prey behavior, we chose several different types of predators: *P. pacificus* strains PS312 and RS5194, a St-only *P. pacificus* mutant TU445 *eud-1(tu445)* (*Ragsdale et al., 2013*), and an isolate of *P. uniformis*, JU1051. *P. pacificus* strain RS5194 is more aggressive than PS312 as characterized by an increased probability of bite per encounter (*Quach and Chalasani, 2022*) so both strains were included in this analysis. The St-only (non-predative) mutant was included to demonstrate whether biting was required for predators to alter *C. elegans* behavior. We also included the cohabiting predator *P. uniformis*. As a more naturalistic predator which has coevolved with *C. elegans*, we wondered how prey response to this predator may differ from *P. pacificus*. *P. uniformis* males and females were considered separately, while only hermaphrodite *P. pacificus* were used. We first tested if short-term predator exposure could alter where eggs were laid by determining the numbers of eggs on and off bacterial lawns in our experimental arenas. These tabulations allowed us to fit a logistic regression model (Materials and methods, *Equations 1 and 2*) that estimated the probability of off lawn egg laying ('P(off)') as a function of time and in interaction with various predators or other conditions. To prevent eggs hatching into L1s, which secrete pheromones that promote lawn leaving (*Scott et al., 2017*) this assay only ran for 6 hr. *C. elegans* in general showed an increase to P(off) over time regardless of predator condition although animals exposed to the aggressive strain *P. pacificus* RS5194 showed slightly higher P(off) at 6 hr compared to mock (*C. elegans* only 0.22, RS5194-exposed animals 0.29, p=0.043) (*Figure 1—figure supplement 2*). We

also observed an increase to P(off) between 3 and 5 hr when exposed to *P. uniformis* females but by 6 hr P(off) in both mock control and *P. uniformis* female-exposed conditions appeared comparable.

We hypothesized that increasing predator exposure time would more greatly increase the probability of off lawn egg laying in predator-exposed animals. We conducted a long-term assay with L4 *C. elegans* and J4 *Pristionchus* instead of adults and stopped the assay after 20 hr of exposure. Juveniles develop into adulthood over the course of the assay (*C. elegans* starts laying eggs approximately 8–10 hr after the L4 stage; *Brenner, 1974*). Thus, as eggs were laid primarily in the latter portion of the 20 hr time period, this limited L1 hatching during the assay. Arenas with *P. pacificus eud-1* mutants showed similar P(off) compared to mock (*C. elegans* only)-exposed animals, while all other *Pristionchus* predators showed pronounced increases to the probability of off lawn egg laying (*Figure 1b–c*). These data indicate that interactions between *eud-1* mutants and prey (secretions, contacts, and others) are unable to alter the locations of *C. elegans* eggs. We confirmed that this change was primarily due to altered egg laying location and not overall changes to the number of eggs (no significant change in egg numbers after predator exposure, *Figure 1—figure supplement 3*). While *P. uniformis* males triggered a similar proportion of *C. elegans* eggs to be laid off lawn (P(off)=0.72) compared to both strains of *P. pacificus* (RS5194 0.74, PS312 0.73), *P. uniformis* females had an intermediate effect (P(off)=0.62). Taken together, these experiments show that *C. elegans* change their location of egg laying away from a lawn occupied by primarily Eu *Pristionchus* capable of biting.

Next, we tested whether *Pristionchus* biting-induced injury was required for the change in *C. elegans* egg location. We used a *C. elegans* reporter strain expressing GFP (green fluorescent protein) under the control of an *nlp-29* promoter. This strain (*Pnlp-29*::GFP) has been shown to increase GFP expression upon wounding the cuticle with a microinjection needle, a laser beam, or fungal infection (*Pujol et al., 2008a*; *Pujol et al., 2008b*). We paired each of the predators tested in our egg location assay with this reporter strain and monitored GFP fluorescence relative to the co-injection marker (*Pcol-12::dsRED*) (*Figure 1c*). We found that both isolates of *P. pacificus* (PS312 and RS5194) were able to increase reporter fluorescence in this reporter strain within 4 hr (*Figure 1—figure supplement 4*). In the 20 hr assay, *C. elegans* exposed to *P. pacificus* RS5194 were killed and could not be measured, but animals exposed to *P. pacificus* PS312 adults showed increased reporter fluorescence (*Figure 1d*). In contrast, the stenostomatus *eud-1* mutant was unable to increase GFP fluorescence even after 20 hr. Notably, neither *P. uniformis* males nor females were able to increase GFP fluorescence in this *Pnlp-29*::GFP reporter strain. However, while it is difficult to confirm biting when the bites are relatively ineffective, *C. elegans* do appear to sense putative bites from *P. uniformis* by exhibiting escape response typical of other aversive stimuli (*Quach and Chalasani, 2022*; *Hilliard et al., 2002*; *Video 1*). It is possible that these bites are causing low level of harm without damaging the cuticle sufficiently to increase expression from the *Pnlp-29*::GFP reporter. We planned to use the *C. elegans* egg location assay for the remainder of our studies in non-fluorescent wildtype (WT) *C. elegans* and so chose a predator that does not lay eggs (*P. uniformis* males) in our assays (*Figure 1e*). Furthermore, failure to elicit a change in *Pnlp-29*::GFP fluorescence also indicated that changes to P(off) when exposed to *P. uniformis* animals in our egg location assay was not the result of extensive injury.

We next tested how the ratio of predators (*P. uniformis* males) to prey (*C. elegans*) affected the location of prey eggs and the expected value of P(off). While maintaining the same arena size and total number of animals (six), we altered the ratio of predators and prey. We found that the presence of even a single predator was able to increase the P(off) and adding additional predators resulted in greater increase to P(off), appearing to asymptote after ≥2 predators in the arena (*Figure 1—figure*

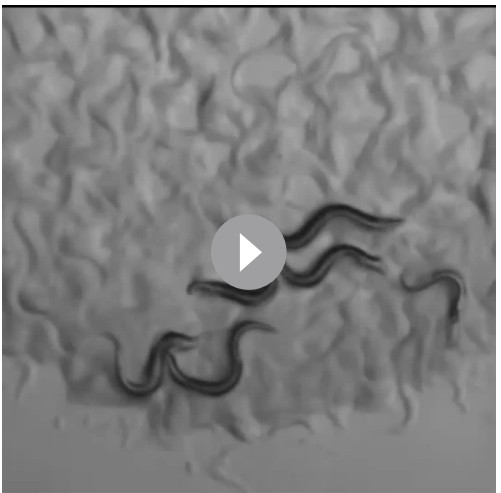

**Video 1.** *C. elegans* avoids *P. uniformis*. Video showing *P. uniformis* and *C. elegans* on a bacterial lawn. *C. elegans* shows rapid avoidance responses to bites from *P. uniformis*. https://elifesciences.org/articles/83957/figures#video1

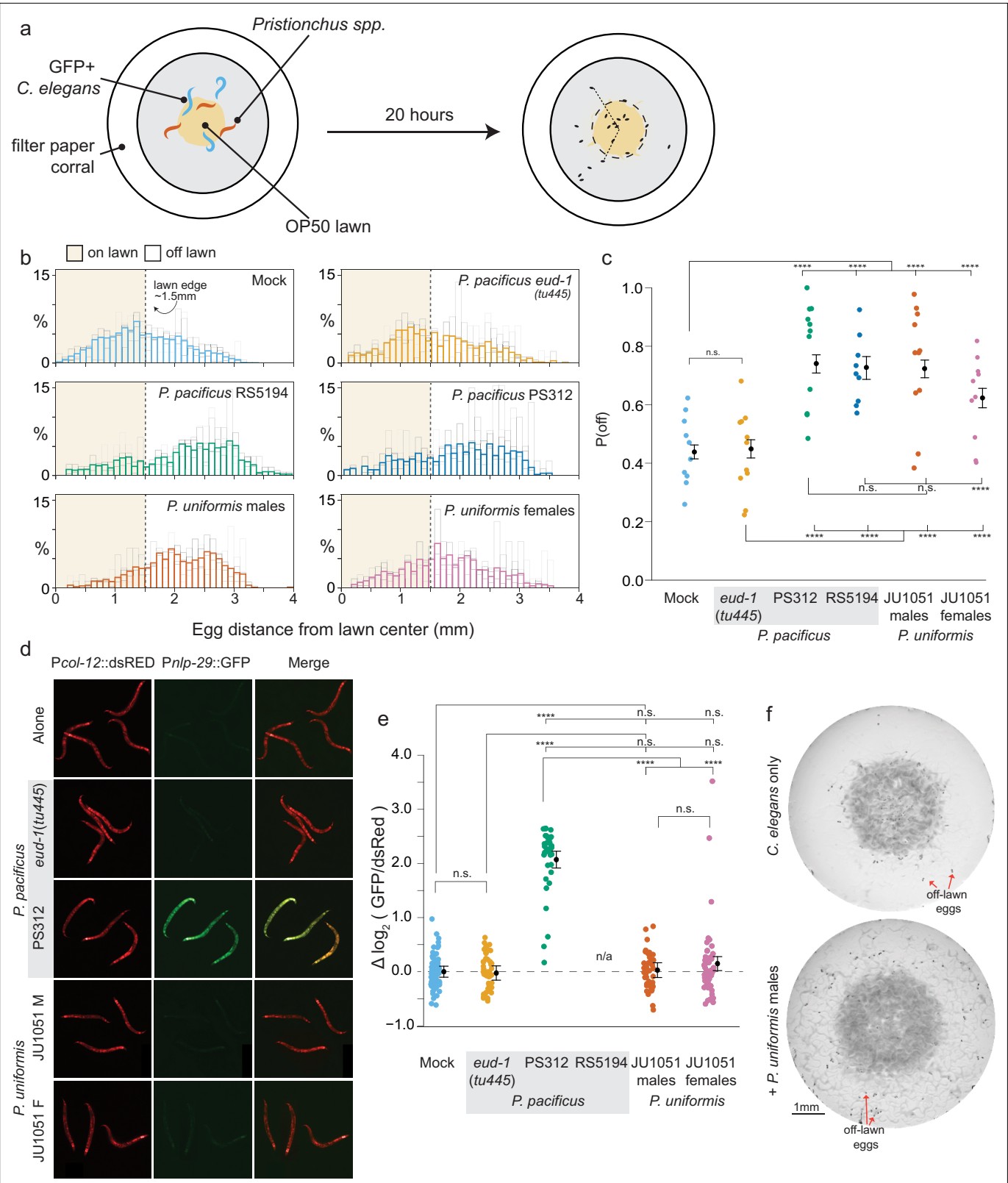

**Figure 1.** Predators influence prey egg location. (**a**) Schematic showing egg location assay setup. Small lawns (approx. 3 mm) in diameter are enclosed in a filter paper corralled arena. Six animals are placed into the arena, three GFP+ *C. elegans* strain CX7389 and three *Pristionchus* spp. (or six *C. elegans* in mock controls). After 20 hr, eggs are visualized and <x,y> positions in the arena are determined. (**b**) Histograms of egg distributions in mock (N=10 arenas) or five predator conditions: *P. pacificus eud-1(tu445)* mutants (N=11 arenas), *P. pacificus* strain PS312 (California isolate) (N=11 arenas),

*Figure 1 continued on next page*

*Figure 1 continued*

*P. pacificus* strain RS5194 (Japan isolate) (N=9 arenas), *P. uniformis* strain JU1051 males (N=11 arenas), and *P. uniformis* JU1051 females (N=10 arenas). Bolded bars show average distribution of egg distance from center (in mm) with faint bars indicating the individual arena distributions. Lawn edge is marked at radial distance approximately 1.5 mm from center. (**c**) Distributions of eggs are quantified as <# eggs off lawn, # eggs on lawn> in each arena and the observed probability of off lawn egg laying (P(off)) is plotted in each condition (# eggs off lawn/total # of eggs, average of ~90 eggs per arena). Statistical analysis was performed by logistic regression in R modeling the [# off, # on] egg counts as a function of predator condition, with significant effects determined by likelihood ratio analysis of deviance in R. Model estimates are overlaid on plots as expected values of P(off) from the logistic model ± 95% confidence intervals. We detected a significant main effect of predator condition ($p < 2.2 \times 10^{-16}$). Post hoc comparisons with correction for multiple testing were computed using the single step multivariate normal procedure in the *multcomp* package in R according to simultaneous method of Hothorn, Brez, and Westfall (*Hothorn et al., 2008*). (**d**) *C. elegans* expressing P*nlp-29*::GFP and a P*col-12*::dsRed co-injection marker paired with various predators after 20 hr. *P. pacificus* RS5194 animals all died following 20 hr of predator exposure. GFP signal was quantified and normalized to dsRed signal for each animal. (**e**) $\log_2$ (GFP/dsRed) signal is shown relative to the mock mean (=0). N=79 mock, 47 *P. pacificus eud-1*(*tu445*), 34 *P. pacificus* PS312, 44 *P. uniformis* JU1051 males, 49 *P. uniformis* JU1051 females. Statistical analysis was performed with ANOVA and we detected a significant main effect of predator condition ($p < 2.2 \times 10^{-16}$). Model estimates are overlaid on plots as mean $\log_2$ normalized fluorescence ± 95% confidence intervals. Post hoc comparisons with correction for multiple testing were performed using the single step multivariate t procedure in the *multcomp* package in R (*Hothorn et al., 2008*). (**f**) Representative images of egg location assay plates after 20 hr of mock (upper) or exposure to *P. uniformis* males (lower). Red arrows indicate example eggs laid off lawn. n.s.=p>0.1, †=p<0.1, *p<0.05, **p<0.01, ***p<0.001, ****p<0.0001.

The online version of this article includes the following source data and figure supplement(s) for figure 1:

**Source data 1.** Egg position data in various predator conditions.

**Source data 2.** P*nlp-29*::GFP and P*col-12*::dsRed data in various predator conditions.

**Figure supplement 1.** Eurystomatous and stenostomatous animals in *P. uniformis* and *P. pacificus.*

**Figure supplement 2.** 6 hour time course of off lawn egg laying with various predators.

**Figure supplement 2—source data 1.** Egg position data in various predator conditions from 1 to 6 hr.

**Figure supplement 3.** Number of eggs laid in arenas after 20 hr of exposure to various predators.

**Figure supplement 4.** Six hour time course of P*nlp-29*::GFP fluorescence with various predators.

**Figure supplement 4—source data 1.** P*nlp-29*::GFP and P*col-12*::dsRed data in various predator conditions from 2 to 6 hr.

**Figure supplement 5.** Different ratios of predator:prey alter the probability of off lawn egg laying.

**Figure supplement 5—source data 1.** Egg position data in various predator:prey ratios.

**Figure supplement 6.** Bacteria pre-conditioned with *P. uniformis* males is not sufficient to alter egg laying behavior.

**Figure supplement 6—source data 1.** Egg position data in arenas with conditioned lawns.

supplement 5a*). These changes to predator:prey ratio did not alter the overall abundance of *C. elegans* eggs (*Figure 1—figure supplement 5b*). These data are consistent with results in our previous study using *P. pacificus* (*Quach and Chalasani, 2022*).

As exposure to *P. uniformis* males did not result in strong injury to *C. elegans* but nevertheless was associated with changes to off lawn egg laying, we wondered whether, rather than biting itself, this phenomenon was due to compounds secreted by the predator. We have previously shown that *P. pacificus* secretions are aversive to *C. elegans* (*Liu et al., 2018*). We tested whether *P. uniformis* was secreting an aversive chemical that drives *C. elegans* away from the bacterial lawn. We conditioned lawns with *P. uniformis* males or sterile *C. elegans* as a control (to simulate changes in lawn caused by animal movement) and tested whether naïve *C. elegans* would alter their egg location on these lawns. We were unable to detect a shift in P(off) as the result of exposure to *P. uniformis*-conditioned lawns (*Figure 1—figure supplement 6a*). We did detect, curiously, an increase to the overall number of *C. elegans* eggs, though this was likely driven by an outlier effect (*Figure 1—figure supplement 6b*). In these assays, *C. elegans* was allowed to lay eggs in the arena for 2 hr. These data suggest that *P. uniformis* males either do not secrete a *C. elegans* aversive signal that can account for the observed predator-induced change to egg location or that *C. elegans* requires substantially longer exposure to such a signal compared to *P. pacificus*.

## Predator-induced changes to off lawn laying are associated with sustained avoidance of the lawn by prey

While *C. elegans* exhibits increased P(off) when occupying a lawn with predators, it may be that *C. elegans* is not truly avoiding the lawn in general, but simply altering its decision about where to lay its eggs. To determine where the prey themselves were located throughout the course of a

predator exposure experiment, we used an imaging setup (WormWatcher) to monitor the locations of mScarlet-expressing *C. elegans* over 20 hr with images of acquired every 4 min (*Figure 2a*). We found that, when exposed to *P. uniformis*, *C. elegans* exhibited a shift in location to just outside the lawn boundary, starting at approximately 5–6 hr (*Figure 2b*). This shift in location was sustained in predator-containing arenas through the remainder of the 20 hr assay, while mock controls exposed only to other *C. elegans* remained mainly within the lawn. Thus, we infer that changes to P(off) observed in our egg location assays is likely a consequence of this sustained avoidance.

## Change in bacterial topography alone contributes to, but does not account for, extent of egg location change

We observed that arenas containing *C. elegans* hermaphrodites and *P. uniformis* males, but not controls, had streaks of bacteria outside the main lawn (*Figure 1e*). Given the duration of our assay, these streaks might represent bacteria that sticks on the *C. elegans* body and gets deposited onto the agar as it exits the lawn. Over the duration of the assay, these streaks grow and are visible to the naked eye by the end of the 20 hr period. We tested whether the presence of streaks outside the main lawn alone could account for the change in egg location. We artificially streaked bacterial lawns at the beginning of the assay and monitored the location of the eggs in these predator-free arenas (*Figure 3b*). Indeed, artificial streaking was able to induce an increase in P(off) nearly threefold, however this response was greater in arenas containing *P. uniformis* (*Figure 3c*). These data showed that the presence of bacteria outside the main lawn can drive egg location change but may not be the only contributor to the decision of where to lay eggs when exposed to predator.

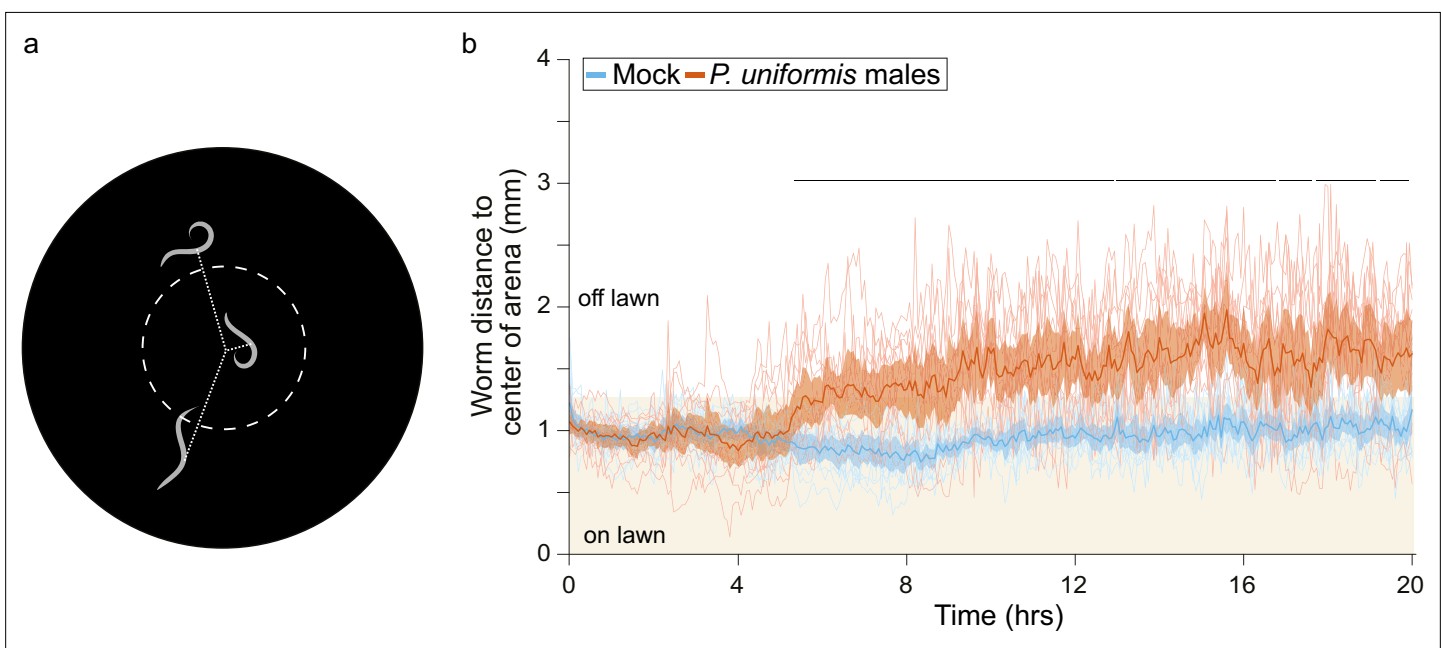

**Figure 2.** *C. elegans* shows sustained avoidance of the lawn when exposed to predator. (**a**) Schematic for WormWatcher experiments for location tracking. Distance of midpoint of fluorescent *C. elegans* (strain ARM112, P*eft-3*::mScarlet) to center of the arena is tracked over 20 hr (15 frames per hour, $t_{resolution}$ = 4 min). (**b**) Worms tracked by WormWatcher included ARM112 strain *C. elegans* in mock (N=12 arenas), or predator (*P. uniformis* males, N=12 arenas), and are plotted as individual traces (thin lines, average position of all worms in an arena, range 2–6 worms per arena, average = 3), representing average distance from center in mm over time. Data were analyzed by non-parametric bootstrap resampling with replacement with 1×10⁵ iterations. Bold lines represent the estimated average distance over time, with shading representing empirically determined 2.5–97.5% quantiles (95% interval) of bootstrap samples. p<0.05 significance can be inferred from regions of lack of overlap of bootstrapped intervals between mock and predator-exposed conditions, identified with lines above traces showing regions of 0% overlap. Regions with 0% overlap account for 71% of all time points, all occurring in the region >5 hr.

The online version of this article includes the following source data for figure 2:

**Source data 1.** WormWatcher tracking data for predator and mock-exposed ARM112 mScarlet expressing *C. elegans*.

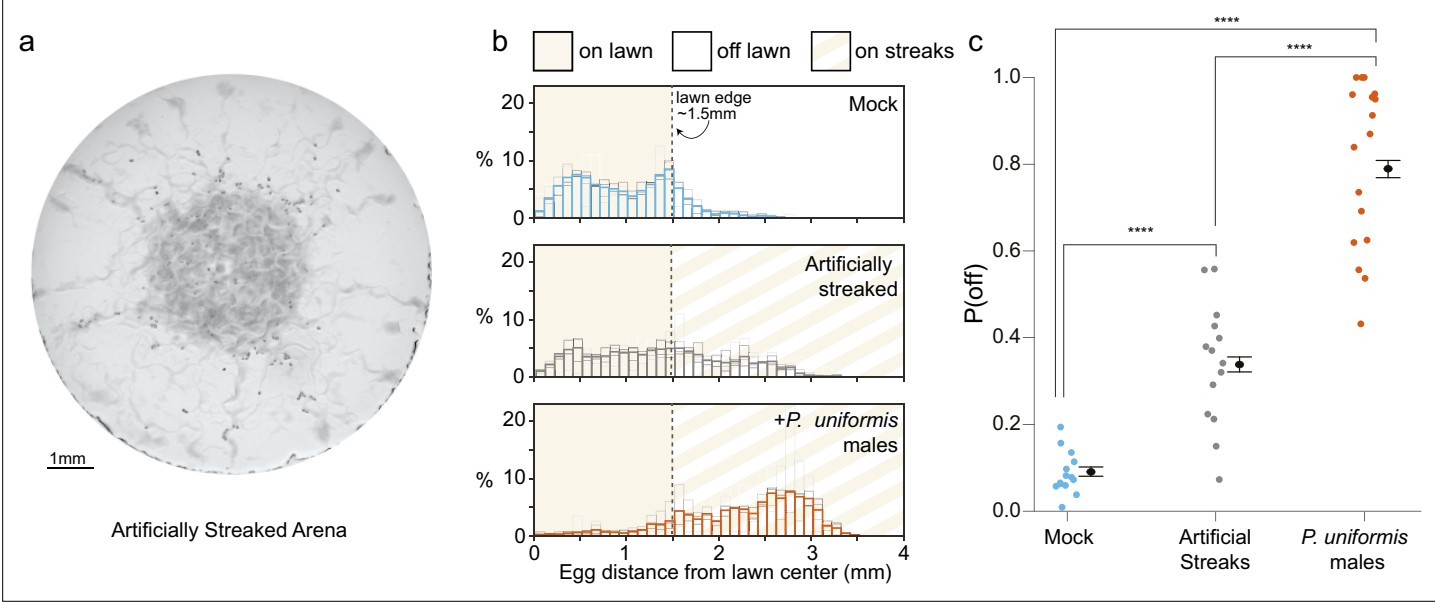

**Figure 3.** Bacterial topography alone does not account for predator associated changed to egg laying location. (**a**) Representative image of an assay plate after 20 hr with artificially streaked lawns. (**b**) Histograms of egg distributions in mock (N=14 arenas), artificially streaked (N=14 arenas), and predator-exposed (N=17 arenas) conditions. Bolded bars show average distribution of egg distance from center (in mm) with faint bars indicating the individual arena distributions. Lawn edge is marked at radial distance approximately 1.5 mm from center. (**c**) Distributions of eggs are quantified as [# off lawn, # on lawn] in each arena as in *Figure 1*, and the observed probability of off lawn laying (P(off)) is plotted in each condition, with data analyzed by logistic regression/analysis of deviance. Overlaid are logistic model estimates of the expected values of P(off)±95% confidence intervals. We detected a significant effect of condition (likelihood ratio $p<2.2 \times 10^{-16}$). Post hoc comparisons with correction for multiple testing were computed using the single step method in the *multcomp* package in R. n.s.=p>0.1, †=p<0.1, *p<0.05, **p<0.01, ***p<0.001, ****p<0.0001.

The online version of this article includes the following source data for figure 3:

**Source data 1.** Egg position data in arenas with and without predator exposure and artificial streaking.

## Egg location change lasts many hours even in the absence of predator

Next, we tested whether changes to egg location persist even in the absence of predators. We 'trained' *C. elegans* prey in our egg location assay setup with *P. uniformis* males for 20 hr and transferred only the prey to a test arena. The position of eggs laid in the test arena was quantified over 6 hr and subjected to the same analyses as our other egg location assays, allowing us to test more nuanced hypotheses about the effect of recent exposure to predator.

We tested animals recently exposed to *P. uniformis* or mock (*C. elegans* only) controls in three environments: completely filled arenas, normal small (~1.5 mm radius) lawn arenas, and arenas with artificial streaks as in *Figure 3*; *Figure 4a*. In a completely filled arena, there is no detectable lawn boundary. Rather than computing a P(off) statistic, we were able to use this arena to estimate predator-induced changes to overall distributional properties of eggs in the absence of a lawn boundary. We looked at the average distance from center eggs laid over 6 hr (*Figure 4b*) as an estimate of the prey's central tendencies, as well as the coefficient of variation of the egg distribution (*Figure 4c*) which estimates changes to the width of the egg distribution that may have been brought about by recent predator exposure. We were unable to detect significant differences due to predator exposure, though we did detect a significant main effect of time on each metric. The average distance of eggs from center decreased over the course of 6 hr, while the coefficient of variation of these distributions tended to increase (*Figure 4c*). The estimated slopes for these effects over time are shown in *Figure 4—source data 3*.

When we tested artificial streaking (*Figure 3*), results suggested both effects of changes to bacterial topology and an interaction with the presence of *P. uniformis* males. We were curious about dynamics of this interaction in the absence of predator. We tested *C. elegans* recently exposed to predator or non-exposed controls in our learning paradigm in arenas containing either a small main lawn or a lawn with artificial streaks, and determined the number of eggs laid at 1–6 hr in independent arenas.

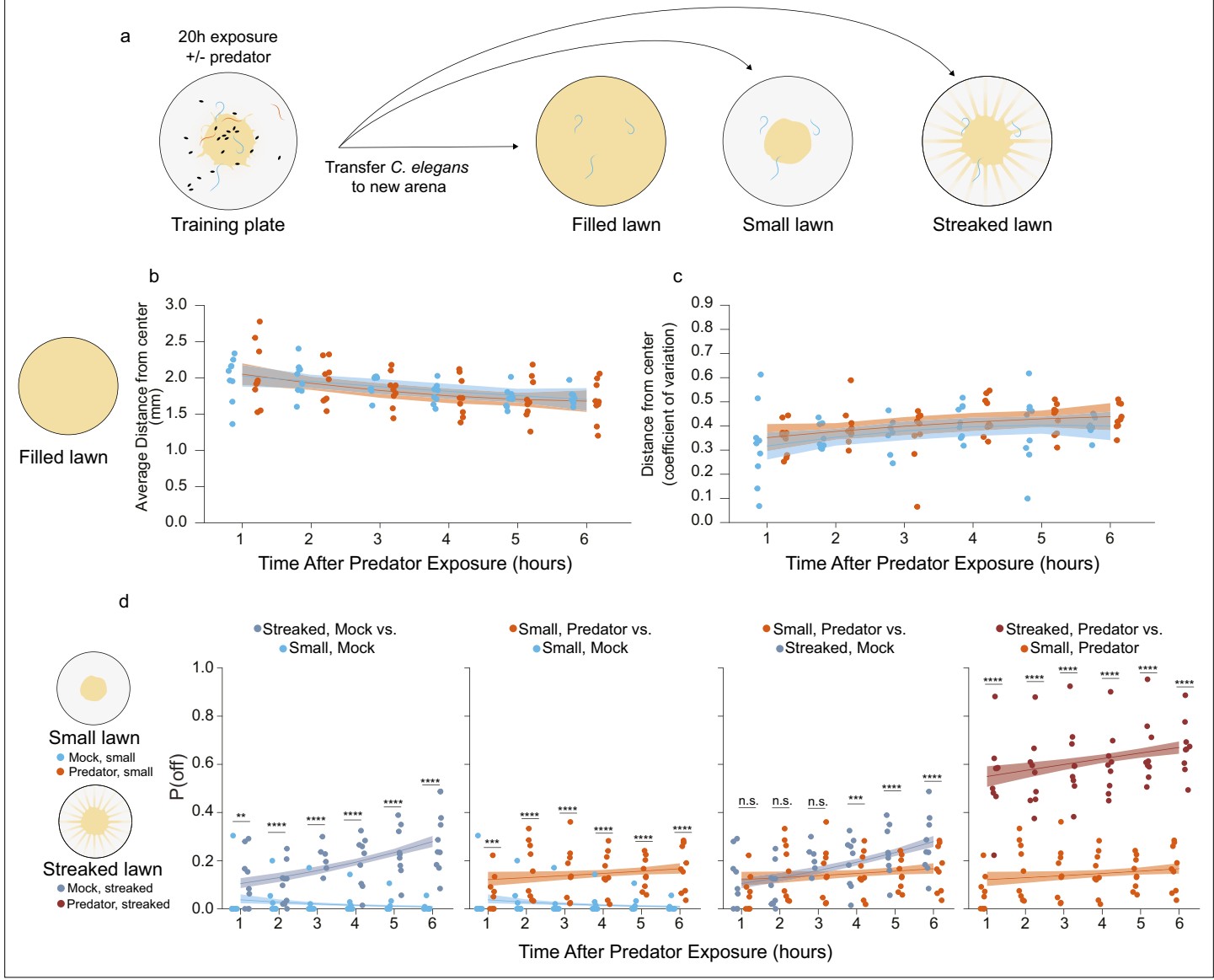

**Figure 4.** Sustained changes to egg laying is observed following prior predator exposure. (**a**) Schematic of egg laying learning assay: after 20 hr of exposure to either mock (*C. elegans* only) or predator condition (*P. uniformis* males), worms are transferred to arenas either completely filled with bacteria or arenas with a normal sized small lawn or a lawn with artificial streaks. In the predator-exposed condition, all three *C. elegans* are transferred, while in the mock condition, three *C. elegans* selected at random from among the six are transferred. (**b**) Analysis of distributional properties of *C. elegans* egg laying for 1–6 hr in arenas completely filled with bacteria after mock (N=8–9 arenas per time point), or predator exposure (N=9 arenas per time point). Plotted are the mean distance from lawn center in mm for each time point and condition. (**c**) Data points represent the coefficient of variation (standard deviation divided by the mean) for egg distances in (**b**) for each time point and for each condition. Data in (**b**) and (**c**) were analyzed by linear regression/ANOVA modeling interactions of time as continuous variable and predator exposure condition. Overlaid on plots are trendlines for each condition from linear models with shading showing 95% confidence intervals. We detected a significant main effect of time on both average distance from center (ANOVA p=$3.0 \times 10^{-6}$) as well as on the dispersal of the eggs measured by the coefficient of variation (ANOVA p=0.0016) but no significant main effect of predator condition (average distance, p=0.51; coefficient of variation, p=0.14) or interaction effects on either variable (average distance, p=0.76; coefficient of variation, p=0.97). (**d**) Analysis of off lawn egg laying in animals exploring small lawns or lawns with artificial streaks after 20 hr of mock or predator exposure. Data points in (**d**) represent observed P(off) in each time point and condition (N=9–12 arenas per time point/ condition). Off lawn egg laying probability was analyzed by logistic regression/analysis of deviance modeling a three-way interaction between time as a continuous variable, lawn type, and predator exposure condition. We detected a significant three-way interaction between these independent variables (likelihood ratio p=$1.5 \times 10^{-7}$). Data in D–G were analyzed together as a single analysis paradigm, however to ease visual understanding of this interaction, pairwise comparisons between conditions are shown in separate panels D–G for: artificially streaked and small lawns for mock-exposed animals, predator vs. mock in small lawns, predator exposure/small lawns compared to the artificially streaked/mock-exposed animals, and finally artificially streaked lawns compared to small lawns for predator-exposed animals. Pairwise comparisons at individual time points between lawn types/

*Figure 4 continued on next page*

*Figure 4 continued*

conditions were computed with correction for multiple testing using the single step method in the *multcomp* package in R. n.s.=p>0.1, †=p<0.1, *p<0.05, **p<0.01, ***p<0.001, ****p<0.0001.

The online version of this article includes the following source data for figure 4:

**Source data 1.** Egg position data in filled arenas after predator exposure.

**Source data 2.** Egg position data in small or artificially streaked arenas after predator exposure.

**Source data 3.** Table of slopes for temporal changes in the distributional properties of eggs after predator exposure in filled arenas.

**Source data 4.** Table of slope for temporal changes to the probability of off lawn egg laying with and without predator exposure, and in arenas with differing bacterial topology.

We found a significant three-way interaction between time, recent predator exposure, and bacterial topology (*Figure 4d*). Animals exposed only to other *C. elegans* and then subsequently laying eggs in test arenas with small unperturbed lawns tended to have very low values of P(off) in general, which decayed negatively over time. By contrast, when tested in arenas with artificial streaks, not only was P(off) increased generally, but showed a positive relationship with off lawn laying increasing over time. When exposed to *P. uniformis* males and tested in arenas with unperturbed lawns, as expected animals did show a potentiation of P(off) and this potentiation to P(off) was comparable to that exhibited by *C. elegans* in the artificially streaked arenas at the early time points. However, in contrast to the temporal dynamics shown by changes to bacterial topology, P(off) was flatter with recent predator exposure across all time points. Finally, combining recent predator exposure and testing on lawns with artificial streaks showed the greatest potentiation to P(off) in general, with a similarly flat response over time. These results suggest that for at least 6 hr there are two separate phenomena: egg laying off the lawn driven by the presence of low concentrations of bacteria at a distance from the main lawn, and egg laying off the lawn driven by recent predator exposure. Predator exposure and artificial streaks together exhibit a combined effect on potentiating P(off) overall which is more than additive. With respect to the time evolution over 6 hr, recent predator exposure appears to trump the effects of bacterial topology, indicated by the relatively flat slopes in predator-exposed animals in either bacterial topological condition. The estimated slopes for these effects over time are shown in *Figure 4—source data 4*.

We wondered whether this elevation to P(off) would persist at even greater periods of time away from predator exposure. We transferred 20 hr exposed *C. elegans* to a rest plate completely filled with food for 1, 2, or 24 hr (*Figure 5a*). We then quantified eggs laid on a test plate containing artificial streaks, as it appeared that artificial streaking of the bacteria was likely to bring about the greatest potentiation of predator-induced changes to P(off). Consistent with the positive slope conditions observed over 6 hr in artificially streaked test arenas (*Figure 4d*), we saw a significant elevation of the baseline level of P(off) at 24 hr in the mock control condition where animals were not exposed to predator (*Figure 5b*). Predator-exposed animals showed elevation to P(off) at all time points including 24 hr, with a flatter relationship over time. This indicates that *C. elegans* are able to 'remember' their recent predator experience for at least 24 hr. However, it is also clear that the baseline probability of off lawn egg laying increases by 24 hr regardless of predator exposure, as exhibited in the mock condition. Thus, we computed changes to the fold change between predator and mock observed at each time point, and defined this as the predator response (see Materials and methods, *Equation 3*). This difference of differences captures the overall magnitude of observed shifts in egg laying behavior associated with the presence of predator. Although our data are not paired, the generalized linear modeling frameworks allows us to compute estimated confidence intervals on this fold change for performing statistical inference. We found that this response was significantly lower at 24 hr than at 1 or 2 hr, as a result of the increase in the baseline P(off) in the mock condition (*Figure 5c*). This indicates that while P(off) remains elevated, *C. elegans* may be beginning to extinguish its 'memory' of recent exposure by 24 hr.

## Biogenic amine signaling regulates off lawn laying behavior

Biogenic amines already are well established as modulators of egg laying behavior in general as well as egg laying during different locomotor modes (*Alkema et al., 2005*; *Cermak et al., 2020*; *Horvitz et al., 1982*). Additionally, biogenic amines are known to modulate behaviors over long time

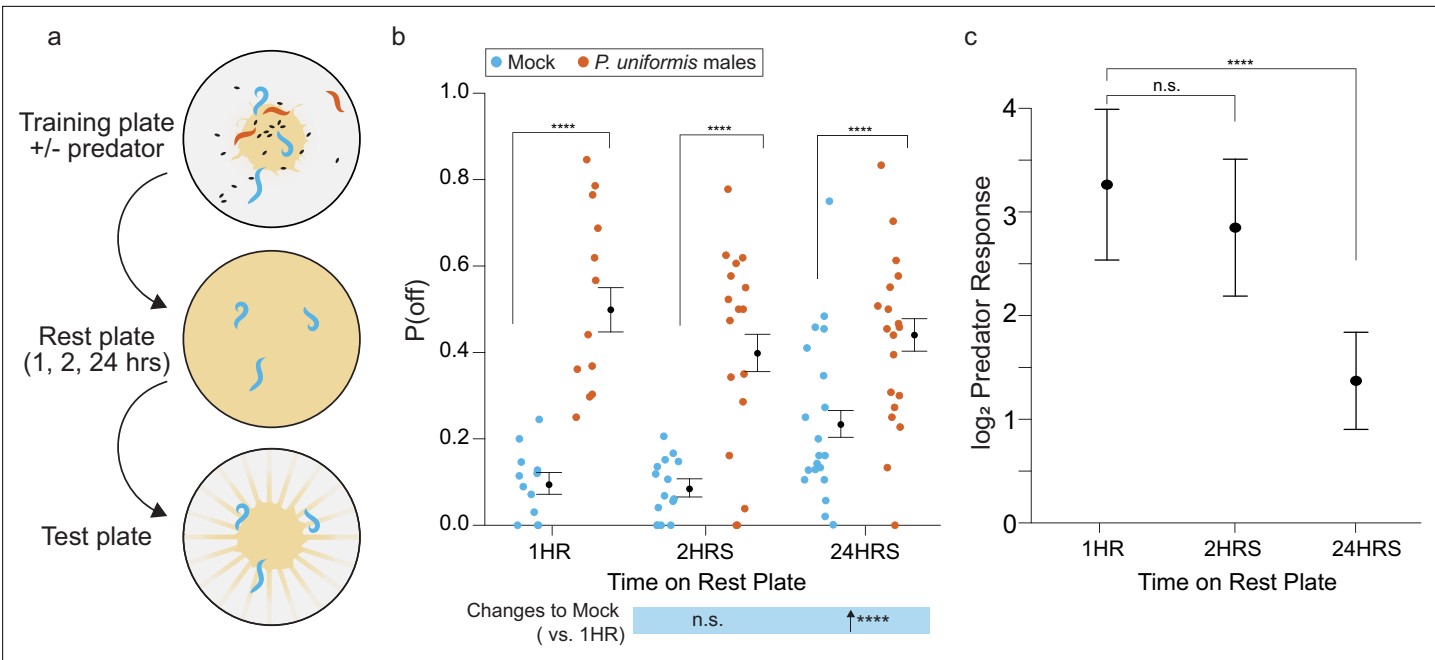

**Figure 5.** Changes to egg laying behavior after predator exposure continue for 24 hr. (**a**) Schematic of egg laying learning assay. *C. elegans* are exposed to mock or predator condition (*P. uniformis* males) for 20 hr and transferred to a rest plate for 1, 2, or 24 hr. After rest, animals are transferred to a test arena containing artificial streaks as in *Figure 4* and positions of laid eggs are determined in order to determine the proportion of eggs laid off and on the lawn. (**b**) Observed P(off) in test arenas is plotted by condition and length of rest period (mock/1 hr N=12 arenas, mock/2 hr N=15 arenas, mock/24 hr N=20 arenas, predator-exposed/1 hr N=12 arenas, predator-exposed/2 hr N=17 arenas, predator-exposed/24 hr N=19 arenas). Data were analyzed by logistic regression/analysis of deviance fitting a two-way interaction of categorical length of rest period (1–24 hr) and mock or predator condition, with expected values of P(off)±95% confidence intervals from logistic model overlaid on plot. We found a significant two-way interaction of rest period length and predator exposure condition (likelihood ratio p=3.4 × 10⁻¹¹). (**c**) Log₂ fold change in computed predator response is plotted for each rest time period, where predator response is defined as the change to the odds ratio of [off lawn/on lawn] egg laying between predator and mock conditions (see Materials and methods, *Equations 1–3*). These are displayed as point estimates with 95% confidence intervals as derived from logistic regression. Post hoc comparisons between conditions, as well as changes to predator response, with correction for multiple testing, were computed using the single step method in the *multcomp* package in R as in previous figures. n.s.=p>0.1, †=p<0.1, *p<0.05, **p<0.01, ***p<0.001, ****p<0.0001.

The online version of this article includes the following source data for figure 5:

**Source data 1.** Egg position data after periods of 1 hr, 2 hr, or 24 hr following predator exposure.

scales (*Chase and Koelle, 2007*), and the change in egg location behavior upon predator exposure appears to last for many hours. We hypothesized that egg laying behavior in response to predator might be subject to modulation by biogenic amines, and therefore tested mutants in genes required for their synthesis. We observed variable changes both to the baseline P(off) probabilities in animals not exposed to predator and to the magnitude of predator exposure response (*Figure 6a–b*). This is consistent with previous studies showing that dopamine and serotonin signaling is required for overall locomotion (*Sawin et al., 2000*; *Flavell et al., 2013*). In general, all mutants were able to show potentiation in P(off) when exposed to predator (*Figure 6a*). Mutants in the *C. elegans* homolog of the mammalian vesicular monoamine transporter, *cat-1* (*Duerr et al., 1999*) exhibited a lower P(off) in the absence of predators. Although *cat-1* mutant animals showed an increase to P(off) with predator exposure, the magnitude of predator response (as in *Figure 5c*) was lower than WT animals (*Figure 6b*). We also found that mutants in *cat-2* (which encodes tyrosine hydroxylase for dopamine synthesis; *Sulston et al., 1975*; *Lints and Emmons, 1999*) and *tph-1* (tryptophan hydroxylase for serotonin synthesis; *Sze et al., 2000*) had similar changes to baseline off lawn egg laying, but nevertheless increase P(off) in the presence of predator. This magnitude of increase was greater than WT in *tph-1* mutants given the very low baseline P(off) in these animals in mock conditions, and the increase in *cat-2*(*e1112*) was similar in fold change magnitude compared to WT, again given their low baseline of P(off) in non-exposed conditions. Mutants in *tbh-1* (which encodes tyramine beta-hydroxylase which converts tyramine into octopamine; *Alkema et al., 2005*) showed a similar baseline of P(off) to WT

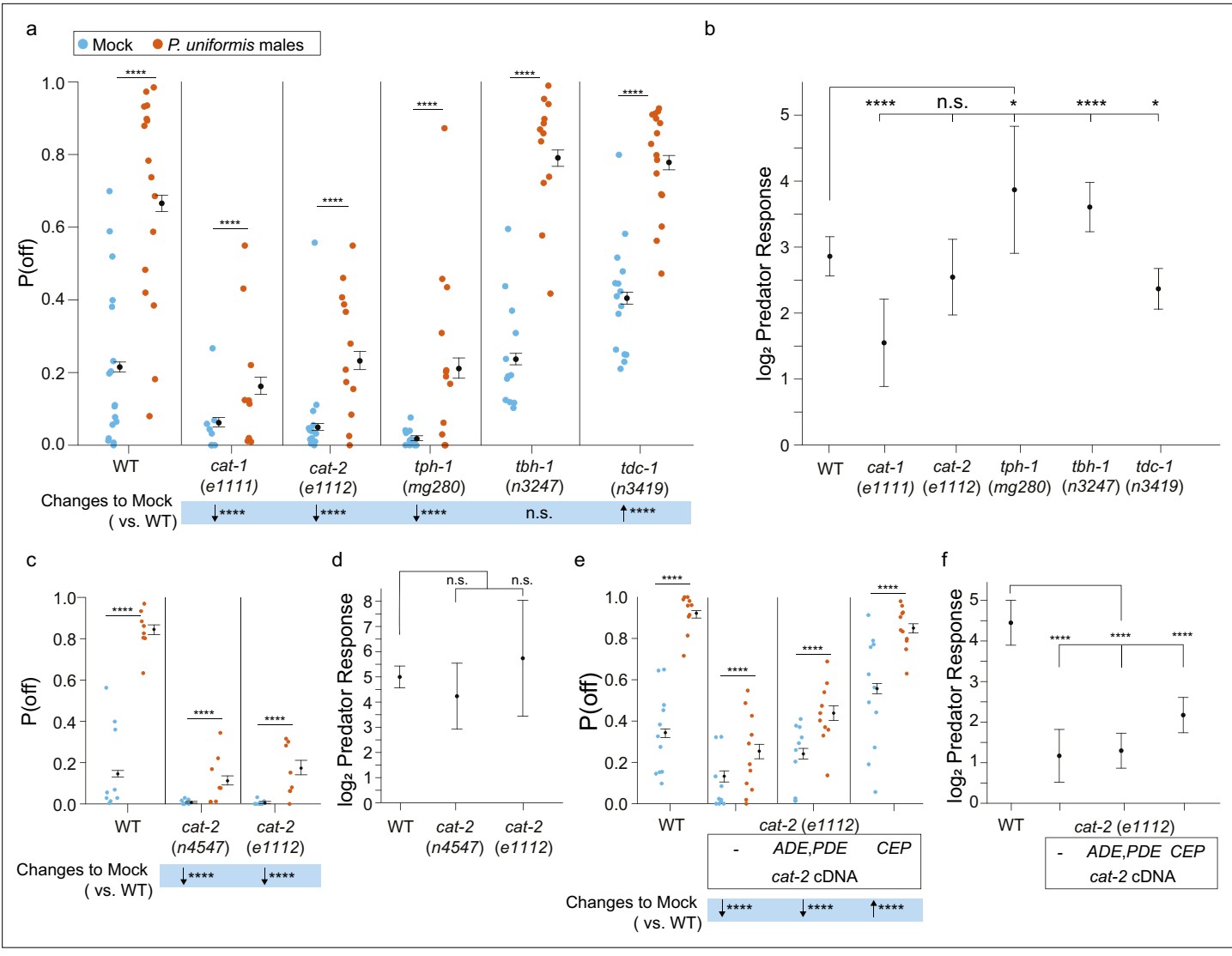

**Figure 6.** Loss of biogenic amine synthesis results in changes to the probability of laying eggs off the bacterial lawn. (**a**) Plotted are observed P(off) data for either mock or predator-exposed arenas in various mutants in biogenic amine synthesis genes (mock: wildtype [WT] N=17 arenas, *cat-1*(*e1111*) N=18, *cat-2*(*e1112*) N=13, *tph-1*(*mg280*) N=12, *tbh-1*(*n3247*) N=12, *tdc-1*(*n3419*) N=15, predator-exposed: WT N=16, *cat-1*(*e1111*) N=9, *cat-2*(*e1112*) N=12, *tph-1*(*mg280*) N=12, *tbh-1*(*n3247*) N=12, *tdc-1*(*n3419*) N=16). Data were analyzed by logistic regression/analysis of deviance fitting a two-way interaction of genotype and predator exposure, with overlaid expected values of P(off) from logistic modeling±95% confidence intervals. We detected a significant two-way interaction of genotype and predator exposure condition (likelihood ratio p<2.2 × 10$^{-16}$). (**b**) Log$_2$ predator response (as in *Figure 5* and Materials and methods, *Equation 3*) is plotted as point estimates with error bars showing 95% confidence intervals across genotypes. (**c**) Observed P(off) data in mock or predator-exposed conditions in WT or two *cat-2* mutant alleles *n4547* and *e1112* (mock N=9 arenas per genotype, predator N=8 arenas per genotype). Data analyzed as in (**a**) with overlaid expected values for P(off) from the logistic model±95% confidence intervals. We failed to detect a significant interaction between genotype and predator condition (likelihood ratio p=0.22) but we were able to detect a main effect of genotype (p<2.2 × 10$^{-6}$) and a main effect of predator exposure (p<2.2 × 10$^{-6}$). (**d**) Log$_2$ predator response across genotypes as in (**b**). (**e**) Observed P(off) in WT or *cat-2*(*e1112*) mutant animals with or without transgenic rescue of *cat-2* cDNA in either ADE/PDE or CEP neurons (mock/WT N=11 arenas, predator/WT N=10 arenas, mock/*cat-2*(*e1112*) N=10 arenas, predator/*cat-2*(*e1112*) N=11 arenas, *cat-2*(*e1112*); *p27::cat-2-sl2-GFP* (ADE/PDE) N=10 arenas for each condition, *cat-2*(*e1112*); *Pdat-1p19::cat-2-sl2-GFP* (CEP) N=11 arenas per condition). Data analyzed as in (**a**, **c**) with overlaid expected values for P(off) from the logistic model±95% confidence intervals. We detected a significant two-way interaction of genotype and predator exposure condition (likelihood ratio p<2.2 × 10$^{-16}$). (**f**) Log$_2$ predator response as described in (**b**) and (**d**) across genotype/transgenic rescue conditions. Post hoc with correction for multiple testing, were computed using the single step method in the *multcomp* package in R as in previous figures. n.s.=p>0.1, †=p<0.1, *p<0.05, **p<0.01, ***p<0.001, ****p<0.0001.

The online version of this article includes the following source data and figure supplement(s) for figure 6:

**Source data 1.** Egg position data in biogenic amine mutants with and without predator exposure.

*Figure 6 continued on next page*

*Figure 6 continued*

**Source data 2.** Egg position data in *cat-2* mutant alleles with and without predator exposure.

**Source data 3.** Egg position data in *cat-2* mutant alleles with and without predator exposure and rescue of *cat-2* cDNA.

**Figure supplement 1.** Mutants in dopamine synthesis and reuptake show varying degrees of predator avoidance.

**Figure supplement 1—source data 1.** WormWatcher tracking data for predator and mock-exposed ARM112 mScarlet expressing *C. elegans* and ARM112 animals with *cat-2*(*e1112*) mutant allele.

**Figure supplement 1—source data 2.** WormWatcher tracking data for predator and mock-exposed ARM112 mScarlet expressing *C. elegans* and ARM112 animals with *dat-1*(*ok157*) mutant allele.

animals and a greater potentiation with predator. Mutants in *tdc-1* (tyrosine decarboxylase, which converts tyrosine into tyramine) showed an elevation of P(off) in mock controls and a slight decrease in fold potentiation in the presence of predator compared to WT animals. Tyramine is known to inhibit egg laying (*Alkema et al., 2005*), however we did not detect significant changes to the number of eggs laid per *C. elegans* animals in *tdc-1* mutants (not shown). Taken together, these data show that loss of biogenic amine neurotransmitters can modify off lawn egg laying behavior, attenuating or even increasing the observed response to predator, though these two phenomena were not so clearly separable. Loss of both dopamine and serotonin neurotransmitters in *cat-1* mutants, however, not only reduced the general probability of off lawn laying but also contributed to the largest blunting of the predator response. We focused our remaining studies on dopaminergic signaling, but future work will investigate the role of serotonin signaling as serotonin has been previously shown to modify egg laying behavior (*Schafer, 2005*; *Schafer, 2006*).

We continued to investigate the consequence of loss of dopamine synthesis by testing a second mutant allele of *cat-2*, *n4547*. Both *cat-2* mutants showed a similar reduction to baseline P(off) in mock conditions (*Figure 6c*), and a similar magnitude of predator response (*Figure 6d*). In *C. elegans* adult hermaphrodites, CAT-2 protein is expressed by eight neurons (four CEPs, two ADEs and PDEs), and dopamine signaling has been previously shown to affect modulation of locomotion as well as learning (*Chase and Koelle, 2007*). Additionally, analysis of the dopamine transporter promoter has identified specific elements that drive expression of transgenes in subsets of these dopaminergic neurons (*Flames and Hobert, 2009*). Using these cell-selective promoter elements, we expressed full-length coding sequence of the *cat-2* cDNA under either CEP- or ADE/PDE-specific promoters. Transgenic rescue in ADE/PDE partially restored baseline P(off) in mock controls (*Figure 6e*), with rescue in CEP neurons resulting in the greatest increase to baseline P(off), even greater than WT levels. In this particular experiment, *cat-2*(*e1112*) mutants did in fact show a blunted predator response even though this metric accounts for the reduced levels of baseline P(off) in the mock condition, and both rescues also show significantly lower predator response compared to WT (*Figure 6f*). This indicates some variability in absolute loss of dopamine synthesis on modulating predator response vs. modulating off lawn egg laying in general. The cohorts of *cat-2* mutants used in *Figure 6a–b*, *Figure 6c–d*, as well as the results shown in *Figure 7* described below, indicate that changes to the underlying probability of laying eggs off the lawn is likely driving any observed effects to predator response. Additionally, differences in promoter strength used to drive expression of *cat-2* may explain why dopaminergic cell types show differing ability to restore baseline P(off). Nevertheless, it is clear that re-expression of CAT-2 protein in either ADE/PDE or CEP is only sufficient to at least partially restore baseline off lawn egg laying behavior.

We also monitored locomotor activity of *cat-2*(*e1112*) animals over the course of 20 hr using the WormWatcher imager. Mutants were still capable of elevating distance from center upon predator exposure. However, there were approximately 40% fewer time points at which mutants differed between mock and predator-exposed conditions as compared to controls (*Figure 6—figure supplement 1a–b*). When computing confidence intervals for the fold increase (change between predator and mock conditions), both *cat-2*(*e1112*) mutants and WT exhibited similar response, though *cat-2* mutants did show lower magnitudes of change at a few time points (*Figure 6—figure supplement 1c*). A mutant in the dopamine reuptake transporter gene *dat-1*, which has increased amounts of dopamine at synapses (*Nass et al., 2005*; *Carvelli et al., 2004*), showed a nearly identical response to WT animals (*Figure 6—figure supplement 1d–f*). Toward the end of the 20 hr observation period, however, *dat-1* mutants in the mock condition began to move away from the lawn, consistent with the

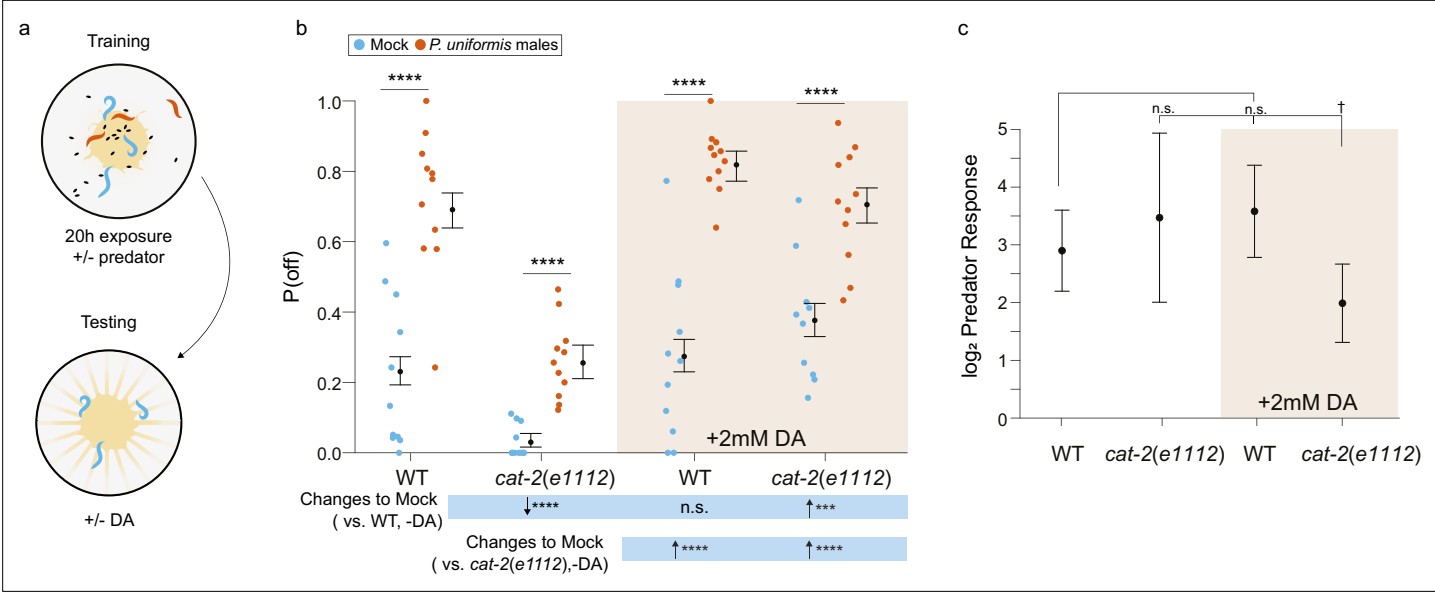

**Figure 7.** Addition of exogenous dopamine rescues egg laying behavior in dopamine synthesis deficient mutants. (**a**) Schematic of the egg laying learning assay. *C. elegans* exposed to either mock or predator condition for 20 hr are transferred to testing arenas containing artificially streaked bacteria with or without the addition of 2 mM dopamine. (**b**) Observed P(off) data are plotted for each genotype, predator exposure, and presence of exogenous dopamine (N=11 arenas all conditions except *cat-2(e1112)*/mock/+3 mM dopamine condition which had N=10 arenas). Data were analyzed as in previous figures by logistic regression/analysis of deviance fitting a three-way interaction of genotype, predator exposure, and dopamine, with overlaid expected values of P(off) from logistic modeling±95% confidence intervals. We detected a significant three-way interaction of genotype, predator exposure, and dopamine (likelihood ratio p=0.0006). (**c**) Log$_2$ predator response±95% confidence intervals (as in *Figures 5–6*, see Materials and methods, *Equation 3*) in each genotype with and without addition of 2 mM dopamine. Post hoc with correction for multiple testing were computed using the single step method in the *multcomp* package in R as in previous figures. n.s.=p>0.1, †=p<0.1, *p<0.05, **p<0.01, ***p<0.001, ****p<0.0001.

The online version of this article includes the following source data for figure 7:

**Source data 1.** Egg position data in *cat-2(e1112)* mutants with and without predator exposure and addition of 3 mM dopamine.

role of excess dopamine in altering locomotion (*Hills et al., 2004*; *Chase and Koelle, 2007*; *Calhoun et al., 2015*). These results suggest that dopamine signaling is required for off lawn exploration and changes in this pathway likely affects both animal position and egg laying distribution.

Next, we hypothesized that adding exogenous dopamine would restore normal egg laying to *cat-2* dopamine deficient mutants. To test our hypothesis, we first exposed WT and *cat-2* mutant *C. elegans* to *P. uniformis* males for 20 hr (training) and then transferred them to a plate with a lawn and artificial streaks (as in *Figures 5–6*) with and without exogenous dopamine (*Figure 7a*). This assay setup avoids exogenous dopamine from altering *P. uniformis* behavior, and leverages our data that prey responses persist for 24 hr even without predators. Previously 2 mM exogenous dopamine has been shown to rescue basal slowing upon encountering food (*Sawin et al., 2000*) and density pattern discrimination of PDMS pillars (*Han et al., 2017*) in *cat-2* mutants. Consistent with our previous results, *cat-2* mutants exhibited reduced off lawn egg laying in both control and predator-exposed conditions (*Figure 7b*). We found that adding 2 mM dopamine restored normal off lawn egg laying in both of these conditions. In the case of *cat-2* mutants, addition of exogenous dopamine restored baseline P(off) to significantly greater levels than in WT, and thus exhibited a net reduction the predator response (*Figure 7c*). Together, these data suggest that dopamine signaling is required for off lawn egg laying in both control and predator-exposed conditions.

## Dopamine receptor signaling alters both baseline and predator-evoked egg laying behavior

Complete loss of dopamine synthesis appeared to primarily affect baseline levels of egg laying activity off the bacterial lawn, so we next explored the roles of the cognate dopamine receptors in modifying this behavior. The *C. elegans* genome encodes at least four dopamine receptors (*dop-1*, *-2*, *-3*, and *-4*) with viable mutants in each (*Chase and Koelle, 2007*). These receptors each have multiple protein

isoforms whose sequence alignments are depicted in *Figure 8a*. *C. elegans* DOP-1 is a homolog of the mammalian D1-like receptors and DOP-2/3 are homologs of mammalian D2-like receptors (*Chase and Koelle, 2007*). DOP-4 is also D1-like, however this receptor belongs to a unique invertebrate family of D1-like including receptors found in *Drosophila melanogaster* and *Apis mellifera* (*Sugiura et al., 2005*). We tested single mutants in each of these four receptors in our egg location assay along with a quadruple mutant that lacked all four receptors. P(off) was increased with exposure to predator in all cases (*Figure 8b*). Complete loss of all four receptors was associated with a trend to reduce the baseline of P(off) in mock controls (p=0.08 after multiple testing correction) and did not show a significant change to the predator response compared to WT (*Figure 8c*), which were similar effects observed when removing dopamine synthesis. Loss of individual receptors had varying results. Loss of *dop-1*, *dop-2*, *dop-3* all elevated baseline P(off) to varying degrees (*Figure 8b*) and showed concomitant reductions to the magnitude of predator-induced fold increases (*Figure 8c*). Thus, loss of single receptors, though able to modulate overall fold change in P(off) when predator was present, still appeared to do so as a consequence of changes to background. Only *dop-4* single mutants show mock condition P(off) not significantly distinct from WT and also showed comparable predator-evoked response.

Since dopamine receptors are known to exist as heteromers (*Perreault et al., 2014*), we analyzed mutants in pairwise combinations. Again, all combinations of two *dop-* mutants showed an elevation of P(off) when predator was present (*Figure 9a*). These combinations also had differing effects on baseline P(off) in mock controls. *dop-1;dop-4* mutants were the most similar to WT. *dop-1;dop-2*, *dop-2;dop-4,* and *dop-2;dop-3* all showed elevation of baseline off lawn egg laying activity relative to WT, and *dop-1;dop-3* and *dop-3;dop-4* showed reductions to baseline P(off). The magnitude of predator response in these mutant combinations is shown ordered from highest to lowest in *Figure 9b*. WT and *dop-3;dop-4* double mutants show the highest fold change increase in P(off) relative to their respective mock controls. All combinations containing *dop-4* rank intermediate with *dop-2;dop-3* and *dop-1;dop-3* ranking lowest. Other than *dop-3;dop-4*, all other combinations showed reduction to predator response relative to WT.

The ranked magnitudes of fold change in predator-exposed conditions suggested that combinations with *dop-4* mutants were intermediate or closer to WT response level, regardless of changes to baseline P(off). To test the hypothesis of the presence or absence of just *dop-4* influencing predator-evoked behavior, we performed an experiment comparing triple mutant animals in *dop-1;dop-2;dop-3* to quadruple mutants of all four receptors (*Figure 9c*). Once again, the quadruple mutant showed reduction to the baseline P(off) in the mock control as in *Figure 8*, however, the triple mutant showed a comparable level of off lawn laying in the mock condition relative to WT. This enabled us to more easily interpret the significant reduction to predator response observed when comparing the triple mutant to the quadruple mutant or WT (*Figure 9f*). Taken together, dopaminergic receptor signaling can affect both baseline off lawn laying activity and predator response, and the specific exclusion of *dop-1*, *dop-2*, and *dop-3* from the assembly of available receptors modulates predator response while maintaining otherwise normal levels of egg laying activity.

## Discussion

In this study, we show that *C. elegans* responds to its predator *P. uniformis* by changing egg laying location relative to a shared food patch. When given the option to find lower density bacterial streaks off of the main lawn, *C. elegans* shift to laying more eggs off the lawn basally consistent with a boost in exploratory behavior when alternate food sources are present (*Figures 3 and 4*). When exposed to predator, *C. elegans* is even more likely to lay its eggs off the lawn (*Figure 3*) when these new food options are available and this effect is greater than either exposure to predator or the presence of these bacterial streaks alone, and persists even in the absence of predator for many hours (*Figures 4 and 5*).

We show that basally and in predator-exposed contexts, a shift to laying eggs off the lawn is modulated by biogenic amine signaling. Biogenic amines like dopamine and serotonin have been previously shown to play a role in driving responses to predator threat in honey bees (*Nouvian et al., 2018*), ants (*Aonuma, 2020*), and fruit flies (*Gibson et al., 2015*). Consistently, we find that loss of dopamine synthesis modulates baseline *C. elegans* egg laying which is consistent with changes to locomotion observed in these mutants (*Chase and Koelle, 2007*). This behavior is rescued by transgenic

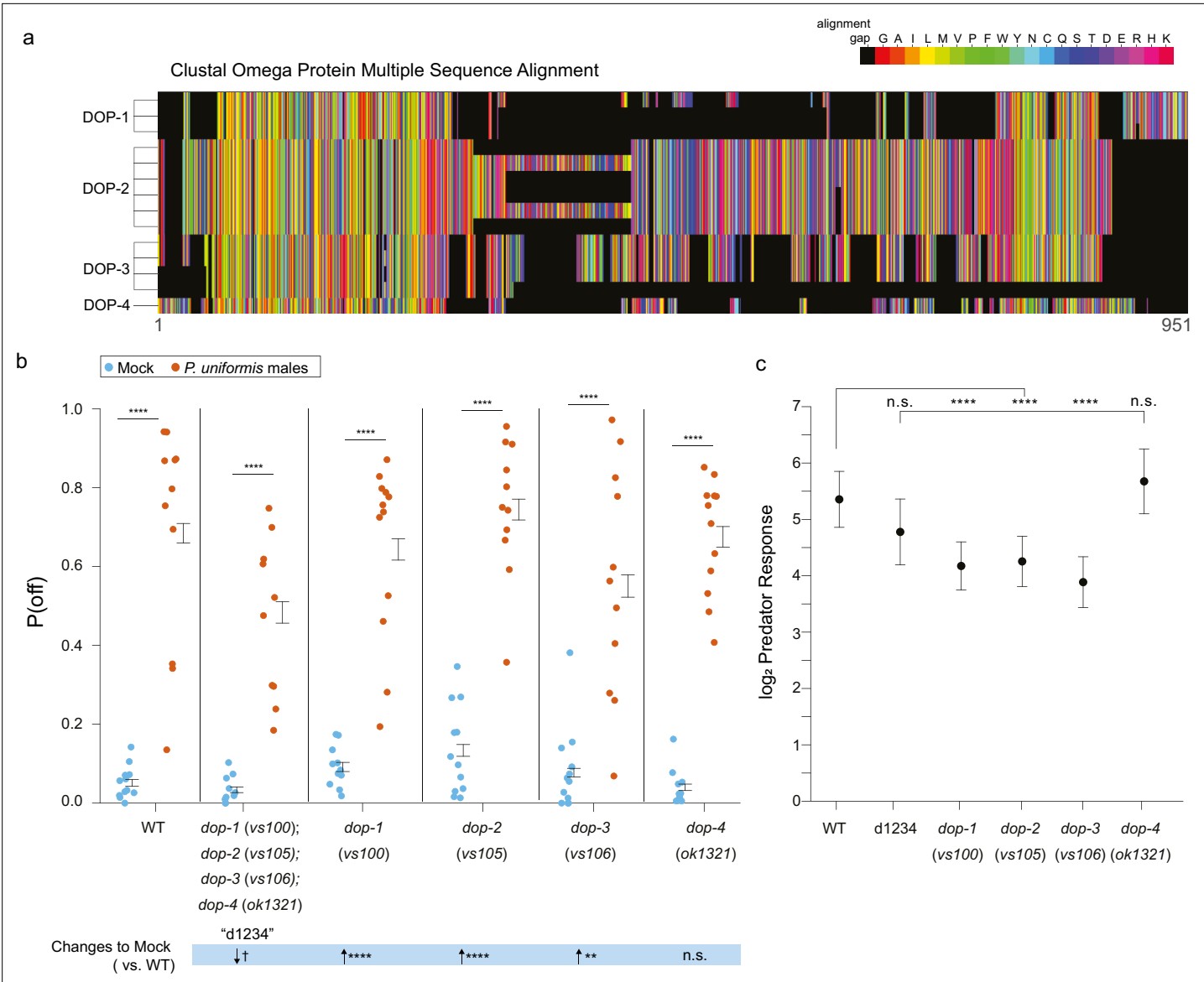

**Figure 8.** Mutations in DOP family dopaminergic receptors influence egg laying behavior with predator exposure. (**a**) CLUSTAL Omega multiple protein sequence alignment of the three isoforms of dopaminergic receptors DOP-1, the six of DOP-2, the three of DOP-3, and DOP-4 are shown visually as a colormap where black squares represent sequence alignment gaps, and amino acids colors are grouped by type (e.g. uncharged, charged). (**b**) Observed P(off) data are shown for the mock and predator-exposed conditions in WT (mock N=12 arenas, predator N=11 arenas), a quadruple mutant for all four receptor genes (N=10/condition), and single receptor mutants *dop-1(vs100)* (N=12/condition), *dop-2(vs105)* (mock N=12, predator N=11), *dop-3(vs106)* (mock N=12, predator N=11), and *dop-4(ok1321)* (mock N=11, predator N=12). Data were analyzed as in previous figures by logistic regression/analysis of deviance fitting a two-way interaction of genotype and predator exposure, with overlaid expected values of P(off) from logistic modeling±95% confidence intervals. We detected a significant two-way interaction of genotype and predator condition (likelihood ratio p<2.2 × 10$^{-6}$). (**c**) Log$_2$ predator response±95% confidence intervals as in *Figures 5–7* (see Materials and methods, *Equation 3*) across genotypes. Post hoc comparisons with correction for multiple testing were computed using the single step method in the multcomp package in R as in previous figures. n.s.=p>0.1, †=p<0.1, *p<0.05, **p<0.01, ***p<0.001, ****p<0.0001.

The online version of this article includes the following source data for figure 8:

**Source data 1.** CLUSTAL multiple protein sequence alignment of DOP receptor amino acid sequences.

**Source data 2.** Egg position data in dopamine receptor mutants with and without predator exposure.

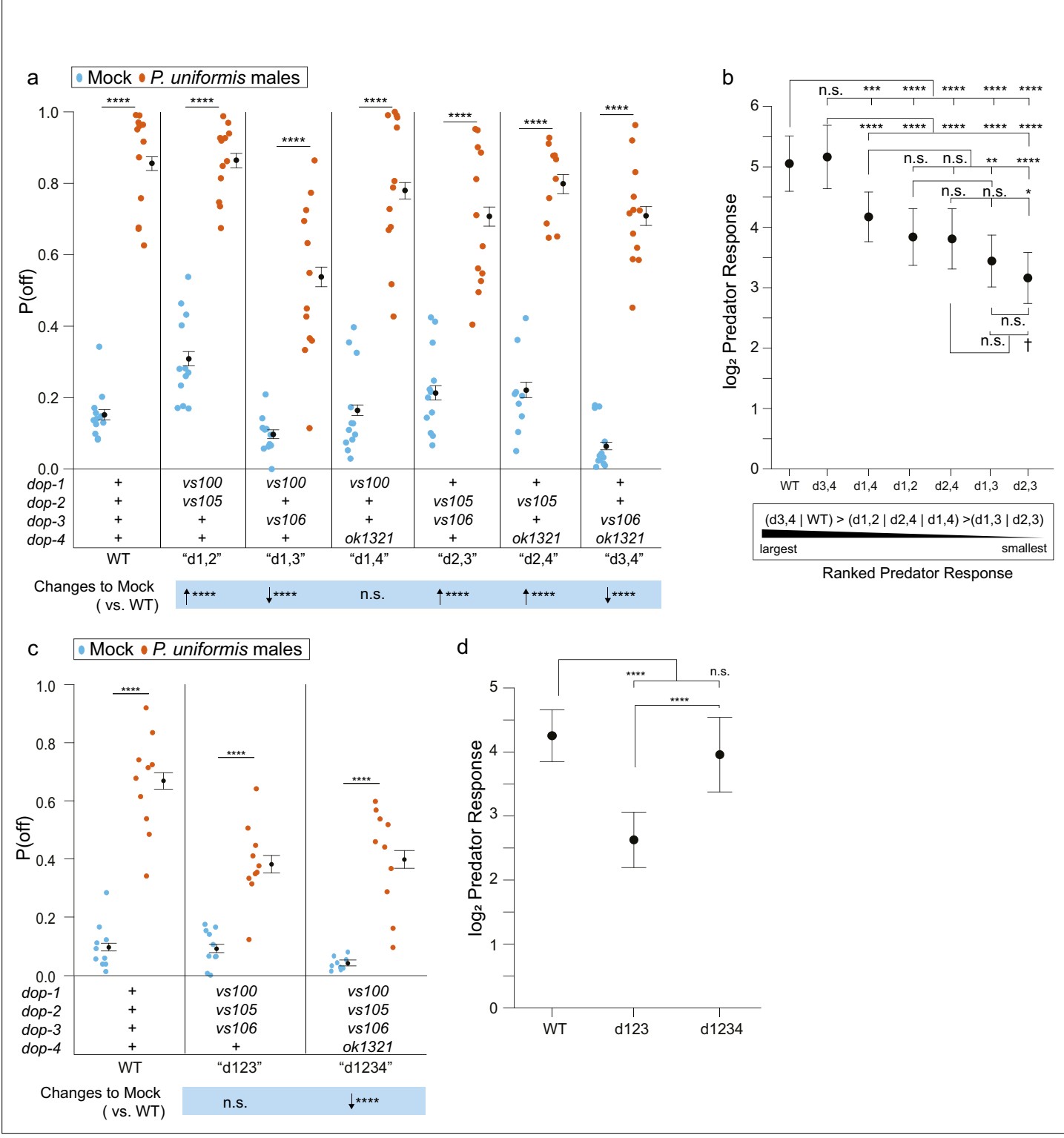

**Figure 9.** Loss of dopaminergic signaling via combinations of DOP receptors is associated with changes to both baseline egg laying behavior and the magnitude of predator response. (**a**) Observed P(off) data are shown for mock and predator-exposed conditions in WT or various pairwise combinations of dopamine receptor mutants (N=12 arenas per condition except mock/*dop-2*(*vs105*);*dop-4*(*ok1321*) N=9, and predator/*dop-2*(*vs105*);*dop-4*(*ok1321*) N=10). Data were analyzed as in previous figures by logistic regression/analysis of deviance fitting a two-way interaction of genotype and predator exposure, with overlaid expected values of P(off) from logistic modeling±95% confidence intervals. We detected a significant two-way interaction of genotype and predator condition (likelihood ratio p<2.2 × 10-6). (**b**) Log₂ predator response±95% confidence intervals as in *Figures 5–8* (see Materials

*Figure 9 continued on next page*

*Figure 9 continued*

and methods, *Equation 3*) across receptor mutant combinations. Below (**b**) is shown a qualitative visualization of the predator response ranked from highest to lowest. (**c**) Observed P(off) data shown for mock and predator-exposed conditions in WT, a triple mutant *dop-1(vs100);dop-2(vs105);dop-3(vs106)*, and a quadruple mutant in all four *dop-* genes (N=10 arenas per condition except mock/quadruple mutant N=9). Data analyzed as in (**a**) with overlaid expected values for P(off)±95% confidence intervals. We detected a significant two-way interaction of genotype and predator condition (likelihood ratio p=2 × 10$^{-13}$). (**d**) Log$_2$ predator response±95% confidence intervals as in (**b**). Post hoc comparisons with correction for multiple testing were computed using the single step method in the multcomp package in R as in previous figures. n.s.=p>0.1, †=p<0.1, *p<0.05, **p<0.01, ***p<0.001, ****p<0.0001.

The online version of this article includes the following source data for figure 9:

**Source data 1.** Egg position data in pairwise combinations of dopamine receptor mutants with and without predator exposure.

**Source data 2.** Egg position data in triple and quadruple dopamine receptor mutants with and without predator exposure.

re-expression of *cat-2* in CEP or ADE/PDE neurons or with the application of exogenous dopamine (*Figures 6 and 7*). Finally, we show that loss of specific combinations of dopaminergic receptors can exhibit effects to the basal rate of off lawn egg laying, but importantly also appear to modulate the magnitude of predator response (*Figures 8 and 9*). Other biogenic amines such as serotonin also appear to exert effects on off lawn egg laying and their contributions to predator-evoked response merit future investigation.

CEP neurons have been previously implicated in learning the size of bacterial lawns. We previously showed that *C. elegans* learns the size of the lawn by using high threshold sensory neurons that detect lawn edges, which in turn signal to CEP neurons to release dopamine. In this paradigm, we speculate that information about lawn size was stored in amount of dopamine released from CEP neurons (*Calhoun et al., 2015*). PDE neurons are involved in increasing egg laying during roaming, and dopamine release can increase the probability of egg laying in the absence of food (*Nass et al., 2005*), so dopamine release from PDE in this predator-prey assay could also encourage egg laying off lawn. While it is the case that effects observed in our transgenic rescue experiments could be due to artifacts of promoter usage, this known division of labor between CEP and PDE could also explain the intermediate levels of rescue to off lawn laying we observe.

We observe a role for multiple dopamine receptors in this prey response to predator threat. The *C. elegans* genome encodes at least four dopamine receptors (*Hobert, 2013*). While DOP-1 and DOP-2/3 are the *C. elegans* homologs of the mammalian D1-like and D2-like receptors respectively, DOP-4 is a D1-like receptor unique to invertebrates (*Sugiura et al., 2005*; *Chase et al., 2004*). We find that the *dop-1; dop-2; dop-3* triple mutant animals have a reduced response to predator threat while maintaining normal off lawn egg laying behavior. Complete loss of all four receptors, or the double loss of *dop-3* and *dop-4*, results in greatest reductions to baseline off lawn laying. Studies in mammals where pharmacology and receptor knockouts have shown that knockouts in D1- and D2-like receptors can have opposing effects on behavior (*Sugiura et al., 2005*; *Kelly et al., 1998*; *Gong et al., 1999*; *McNamara et al., 2003*). Here, we show that specific combinations of receptors can exert varying effects. While we did not identify the site of action of these receptors, we suggest that the combined action of DOP-1, -2, and -3 receptors act downstream of dopamine release to alter prey egg location in predator-exposed animals.

## Ideas and speculation

We speculate that responses by *C. elegans* to predator exposure fit within the broader context of prey refuge, wherein a prey adopts a strategy to reduce predation risk. The prey refuge brings with it the potential cost of decreased feeding opportunities, which is weighed against the benefit of minimal harm induced by the predator (*Sih, 1987*). This theoretical framework is consistent with the interactions we find between predator and changes to bacterial topology. Predator-exposed *C. elegans* shift egg to streaks away from a central lawn, and this strategy may lower the encounter probability with *Pristionchus* thus minimizing risk to the prey. This is especially so given the observations we have previously made that *Pristionchus* predators prefer to patrol a main lawn when it is available, thus leaving refuges open for exploitation by *C. elegans* (*Quach and Chalasani, 2022*). However, such refuges afforded by these streaks may have detriment to *C. elegans* fitness due to their lower density of available food, and longer term monitoring of health and fitness of prey in these conditions has yet to be tested. Variations to the number of available refuges for fleeing prey, as well as their local

food density and quality, can be modified in the future to gain a better appreciation for this intriguing model of prey risk minimization strategy in *C. elegans*.

Dopamine has been shown to affect multiple *C. elegans* behaviors including locomotion, foraging, and learning (*Hills et al., 2004*; *Calhoun et al., 2015*). For example, we previously showed that this pathway is required for learning the size of a bacterial lawn and then driving a search strategy when removed from that lawn (*Calhoun et al., 2015*). Furthermore, dopamine has been shown to promote egg laying when animals roam (*Cermak et al., 2020*). This may explain the interaction effects we observe between predator exposure and artificially streaking bacteria. The combination of these inputs may motivate a roaming program, which continues to promote egg laying at a distance, explaining the large boost in P(off) observed in *Figures 3c and 4d*. Our video tracking data (*Figure 6—figure supplement 1*) suggests that *cat-2(e1112)* mutants are able to avoid the predator at least some of the time at perhaps an attenuated magnitude of response. However, despite this, they very rarely lay eggs off the lawn at all with P(off) values as low as 0.004 and as high as 0.13 across all experiments. Given that loss of dopamine synthesis appears to suppress P(off) and addition of exogenous dopamine restores this baseline (*Figures 6–7*), this is consistent with the hypothesis that dopamine is important in modulating egg laying while roaming. Even when *cat-2* mutants are straying from the lawn, they are still by and large laying eggs on the lawn.

It is curious that in combination, loss of signaling via the DOP-1;DOP-2;DOP-3 receptors modulates predator response without modulating baseline P(off), while additional loss of DOP-4 modulates the baseline. This suggests that potentially the route through which egg laying while roaming is altered requires DOP-4, while predator response proceeds through signaling via the other receptors. Double mutant combinations in our data however are complex with both effects to magnitude of predator response and baseline. These data nevertheless stratify combinations with DOP-3 and either DOP-1 or DOP-2 as showing the most attenuated predator responses (*Figure 9b*). In Chase and colleagues' work identifying DOP-3, triple mutants in *dop-1;dop-2;dop-3* show attenuated basal slowing response in the presence of food, but nevertheless show normal dopamine-dependent paralysis, and this is also exhibited by *dop-1;dop-3* double mutants (*Chase et al., 2004*). It may be that predator-evoked changes to shifting egg locations is linked to lawn edge detection. As *dop-1;dop-2;dop-3* mutants in our work here show similar background levels of P(off) in the mock control condition, this may indicate that lawn edge detection is not the driving force in basal off lawn egg laying. However, when *C. elegans* learns to associate the main lawn with an aversive stimulus such as predator threat, then proper detection of the lawn edge would be crucial to avoiding it. However, when *dop-4* is also mutated, it may be that this mimics loss of dopaminergic signaling observed in dopamine synthesis mutants, which in turn serves to modulate the baseline off lawn laying rate.

In the future, the role of serotonin should be further investigated. Serotonin has been shown to modulate dopamine-dependent behaviors. For example, while dopamine signaling is required for basal slowing when encountering a lawn of food, serotonin can enhance the slowing response if the animal is starved (*Sawin et al., 2000*). Thus, dopamine modulates basal behavior while serotonin modulates it in an experience-dependent manner. Whether serotonin acts in a similar manner in this assay is yet to be investigated.

In summary, after predator exposure, *C. elegans* lays eggs in areas of high food variability that still have some food, rather than laying eggs in a dense food patch inhabited and preferred by predators. Loss of dopamine synthesis alters baseline egg laying activity restored by exogenous dopamine, while nuanced combinations of dopaminergic receptors exert effects on specific predator-evoked response. This study lays the foundation for studying prey behavior in *C. elegans*. Future studies can use this system to interrogate the impact of various neurotransmitter signaling pathways on *C. elegans* feeding, reproductive, and general exploration strategies modified by experience.

## Materials and methods
### *C. elegans* and *Pristionchus* spp. strains
Nematode strains used in this study are shown in the following table. Mutant crosses generated for this study in the table below available upon request (CGC = Caenorhabditis Genetics Center).

| Strain Name | Source | Genotype | Figure | Notes |
|---|---|---|---|---|
| N2 | CGC | Wildtype | *Figure 1—figure supplements 5–6, Figures 2–9* | |
| CX7389 | *Quach and Chalasani, 2022* | kyIs392 [Pstr-2::GFP::rab-3; Pttx-3::lin-10::dsRed; Pelt-2::GFP] | *Figure 1b–c, Figure 1—figure supplements 2 and 3* | Fluorescent eggs |
| CZ6326 | *Pujol et al., 2008a; Pujol et al., 2008b* | frIs7 [nlp-29p::GFP+col-12p::DsRed] IV | *Figure 1d–e, Figure 1—figure supplement 4* | Injury reporter |
| ARM112 | CGC | wamSi112 [eft-3p::mScarlet::unc-54 3'UTR+Cbr-unc-119(+)] II; unc-119(ed3) III | *Figure 2, Figure 6—figure supplement 1* | WT whole-body fluorescent strain |
| IV983 | This study | cat-2(e1112) wamSi112[eftp-3::mScarlet::unc-54 3'UTR+Cbr-unc-119(+)] II | *Figure 6—figure supplement 1* | cat-2 mutant crossed into ARM112 background |
| IV988 | This study | wamSi112[eftp-3::mScarlet::unc-54 3'UTR+Cbr-unc-119(+)] II; dat-1(ok157) III | *Figure 6—figure supplement 1* | dat-1 mutant crossed in ARM112 background |
| CB1111 | CGC | cat-1(e1111) X | *Figure 6a–b* | |
| CB1112 | CGC | cat-2(e1112) II | *Figures 6, 7* | |
| MT13113 | CGC | tdc-1(n3419) II | *Figure 6a–b* | |
| MT15434 | CGC | tph-1(mg280) II | *Figure 6a–b* | |
| MT9455 | CGC | tbh-1(n3247) X | *Figure 6a–b* | |
| MT15620 | CGC | cat-2(n4547) II | *Figure 6c–d* | |
| IV111 | *Calhoun et al., 2015* | cat-2(e1112) II; ueEx51 [p27::cat-2-sl2-GFP; Pelt-2::GFP] | *Figure 6e–f* | |
| IV552 | *Calhoun et al., 2015* | cat-2(e1112) II; ueEx355 [Pdat-1p19::cat-2-sl2-GFP; Pelt-2::GFP] | *Figure 6e–f* | |
| LX645 | CGC | dop-1(vs100) X | *Figure 8b–c* | |
| LX702 | CGC | dop-2(vs105) V | *Figure 8b–c* | |
| LX703 | CGC | dop-3(vs106) X | *Figure 8b–c* | |
| RB1254 | CGC | C52B11.3(ok1321) X | *Figure 8b–c* | |
| LX705 | CGC | dop-1(vs100);dop-3(vs106) X | *Figure 9a–b* | |
| LX706 | CGC | dop-2(vs105) V; dop-1(vs100) X | *Figure 9a–b* | |
| IV984 | This study | dop-4(ok1321);dop-1(vs100) X | *Figure 9a–b* | |
| IV985 | This study | dop-2(vs105) V;dop-4(ok1321) X | *Figure 9a–b* | |
| IV986 | This study | dop-4(ok1321) dop-3(vs106) X | *Figure 9a–b* | |
| LX734 | CGC | dop-2(vs105) V; dop-1(vs100); dop-3(vs106) X. | *Figure 9c–d* | |
| CF2805 | CGC | dop-2(vs105) V; dop-4(ok1321) dop-1(vs100) dop-3(vs106) X | *Figure 8b–c, Figure 9c–d* | |
| CF1903 | CGC | glp-1(e2144) III | | |
| JU1051 | From Marie-Anne Félix (*Félix et al., 2013*) | P. uniformis wild isolate | *Figures 1–9* and figure supplements | |
| PS312 | From Ralf J Sommer (*Click et al., 2009*) | P. pacificus California isolate | *Figure 1b–e, Figure 1—figure supplement 1, Figure 1—figure supplements 1–4* | |
| RS5194 | From Ralf J Sommer (*Click et al., 2009*) | P. pacificus Japanese isolate | *Figure 1b–e, Figure 1—figure supplements 2–4* | |
| TU445 | From Ralf J Sommer (*Ragsdale et al., 2013*) | P. pacificus eud-1(tu445) X | *Figure 1b–e, Figure 1—figure supplements 2–4* | |

## Nematode growth

Nematode strains were maintained at 20°C on 6 cm Petri plates containing Nematode Growth Medium (NGM) seeded with *Escherichia coli* OP50 bacteria (CGC) as food (*Brenner, 1974*).

## Egg location assay

Assay plates are created by spotting 0.5 µl of OP50 liquid culture (OD600=0.5) on 35 mm standard NGM plates (*Brenner, 1974*). The bacterial lawns are allowed to grow at 20°C for 30 hr, then stored

for up to 1 month at 4°C. Whatman filter paper with ¼" punch forms the 'corral' and encircles the lawn, allowing approximately 1.5 mm of clean agar in between the lawn edge and the corral edge. All animals are allowed to crawl on a clean section of agar to clean them of bacteria and picked to the assay plate using a sanitized eyelash, placed next to the lawn on a clean area of agar. Three predators are picked first, staged by overall size and pigment development as J4s. Then, three *C. elegans* L4s are picked to the assay plate. The animals are allowed to interact for a determined amount of time, 20 hr for an overnight assay, at 20°C. For short-term exposure (6 hr and under), gravid *C. elegans* adults and adult predators are used by picking L4s or J4s the day before to plates with plenty of food. The juveniles are allowed to grow overnight into adulthood and then used in the same assay setup. After their interaction, corrals and all adults are removed from the plate and the area inside the corral is imaged using an AxioZoom V16 (ZEISS).

For the streaked lawn variation, streaks are formed by gently dragging a sanitized eyelash through the center of the lawn in radial streaks 10 times, followed by two concentric circular streaks halfway between the lawn and the corral edge. The streaked lawn is then used immediately.

## Injury assay

Injury assays are set up in the same way as the egg location assays, using a *C. elegans* strain containing the array *frIs7* [*Pnlp-29*::GFP+*Pcol-12::DsRed*]. After the set interaction time, worms are immobilized by placing the plates on ice and imaged on an AxioZoom V16 (ZEISS) within 1 hr, with exposure times kept constant for fluorescence imaging (25 ms).

## Learning assay

*C. elegans* are trained using the 20 hr egg location assay. At the same time as the animals used for training are transferred to their assay plates, test plates are set up. Three types of test plates are used: a filled lawn (10 µl of OP50 [OD600=0.5]), a streaked lawn (same as the streaked lawn variant of the egg location assay), and a small lawn (same as the original assay plate). The training plates with animals on them and the test plates are incubated at 20°C for 20 hr, during which the *C. elegans* is exposed to JU1051 males and the smears on the test plates are allowed to grow. (The bacteria on the other test plates are also allowed to grow at this time so that the bacteria are at a similar metabolic state and density across test plates, and streaks are already present.) Filter paper corrals like those used in the egg location assay are centered over the test plate lawns.

After the *C. elegans* are incubated in their training conditions for 20 hr, they are carefully removed with an eyelash pick from their training plates to a clean section of an NGM plate. The animal is allowed to crawl for a few seconds to remove bacteria and then picked to a test plate halfway between the central lawn and the corral edge. For the filled test lawns, the animals are placed in an equivalent position relative to the corral edge. The test plates are then imaged every hour on an AxioZoom V16 for 6 hr. For the variant including rest plates, the *C. elegans* are picked from their training plate to rest plates ('filled lawn' plates) for the set rest time. They are then transferred to a streaked lawn test plate and egg locations are observed after 2 hr. In learning experiments, all three *C. elegans* in predator-exposed conditions are transferred to a rest plate or test arena. In mock controls, where there are six *C. elegans* present, three *C. elegans are* selected randomly for transfer.

## Exogenous dopamine assay

When adding exogenous dopamine to the learning assay, a 200 mM stock of dopamine hydrochloride (Code 122000100 Lot: A0427132, CAS: 62-31-7, Acros Organics) in water was prepared. Two hours before the trained worms needed to be transferred to the test plates, 50 µl of the dopamine stock or water as a control was gently applied onto the streaked lawn test plate. The plates were allowed to diffuse and dry with the lids off for 2 hr, at which time the trained worms were transferred to the test plates. The trained worms were allowed to lay eggs for 2 hr before their plates were imaged.

## Egg location image quantification

Egg location images are quantified in FIJI with the experimenter blinded to the condition by randomizing the file order and obscuring the filenames (using the Filename_Randomizer macro found at https://imagej.nih.gov/ij/macros/Filename_Randomizer.txt). Eggs are manually selected with the multipoint tool and lawns are selected as circles. Distances from each egg from to the lawn edge are

calculated in Python. All assays are performed with their relevant controls over at least 3 separate days.

## WormWatcher assays

Assays conducted in the WormWatcher (Tau Scientific Instruments, West Berlin, NJ, USA) were performed on a single 6 cm 2.5% agar NGM plate in a 12-arena setup. The 12-well corral was created by cutting a 3×4 array of ¼" circles into a plastic sheet using a Cricut machine. OP50 was spotted in the 3×4 pattern using the same concentration and allowed to grow for the same amount of time as in the egg location assay although in this setup lawn radii averaged approximately 1.2 mm as compared to 1.5 mm in other assays. The increased agar percentage on the WormWatcher plates helped prevent worms from escaping under the plastic edges of the corral.

The assays were set up like the egg location assays, with three L4 *C. elegans* and three J4 JU1051 males or six L4 *C. elegans* in control arenas. The positions of predator-containing and/or mutant-containing arenas were alternated on different assay days. The WormWatcher was set to acquire fluorescent frames with a green LED excitation light every 4 min. A reference darkfield image was acquired before and after every experiment to reference the positions of the arenas and the size and positions of the lawns. After the experiment was completed, each arena was inspected and imaged to determine whether any worms escaped away or into it. Custom code was written to segment the *C. elegans* and arenas in each position and the median distance from the lawn edge to the midpoint of each worm body per well was recorded. Data from arenas were discarded if two worms had escaped from an arena, or if a *P. uniformis* was seen in a control arena.

## *Pristionchus* mouthform analysis

*Pristionchus* mouthform analysis was performed as reported in *Werner et al., 2017*. Briefly, *Pristionchus* were egg-prepped via bleaching and eggs were cultured either on standard solid NGM plates or in liquid culture. After eggs reached adulthood, they were immobilized on agarose slides with sodium azide. The slides of different strains from different culture conditions were mixed and their labels obscured while they were observed. The slides were scored as either Eu (wide mouth, two teeth) or St (narrow mouth, one tooth) while the experimenter was blinded to strain.

## Statistical methods

### Replication

All data points represented as scatter points in plots, individual sample traces in *Figure 2*, as well as *Figure 6—figure supplement 1*, as well as coordinates of egg positions in source data files represent biologically distinct samples arising from independent animals, and not technical replication (repeated measurements on the same biological sample). Specific numbers of replicates per condition in assays are displayed in figure legends.

### Egg location data

For egg location assays the number of eggs on and off the bacterial lawn were quantified from images. The probability of laying eggs off the lawn is a bounded variable between 0 and 1 best represented by binomial probability. Thus, tabulated egg data as numbers of eggs off and on the lawn were analyzed via binomial generalized linear models (logistic regression) in R using the glm function to fit one-, two-, or three-way interactions between independent variables (*R Development Core Team, 2009*). These models are fit using the logit link function (*Equation 1*):

$$\log \left( \frac{\mathrm{P(off)}}{1 - \mathrm{P(off)}} \right) = \ \mathbf{X} \cdot \beta \tag{1}$$

where P(off) is the expected probability of off lawn egg laying, X is the design matrix of categorical or continuous predictors, and β is the vector of fitted coefficients. The quantity $\frac{\mathrm{P(off)}}{1 - \mathrm{P(off)}}$ is the 'odds ratio' of laying eggs off the lawn, and thus the logit is the logarithmic scale odds ratio. Changes to the log odds ratio can be interpreted as changes to odds of laying eggs off lawn vs. on lawn. The expected probability P(off) under different conditions and associated confidence intervals can be determined from exponentiation of logit scale quantities using the inverse logit function (*Equation 2*):

$$P\left(\text{off}\right) = \frac{e^{X \cdot \beta}}{1+e^{X \cdot \beta}} \tag{2}$$

These estimates for the expected value of P(off) with its associated 95% confidence interval were used for overlaying on plots. Omnibus effects in the data were determined by likelihood ratio tests/analysis of deviance using the ANOVA function in the car package in R (*Fox and Weisberg, 2018*). Where significant main effects or interactions were detected, post hoc linear hypotheses included both comparisons between groups as well as higher order comparisons of magnitudes of change were computed (as in *Figures 5–9*, e.g. the change to the magnitude of change between predator and mock conditions between genotypes). The 'predator response' in *Figures 5–9* is specifically defined as the change to the expected value of the log odds ratio (*Equation 1*) between mock and predator conditions (*Equation 3*):

$$\log\left(\textit{Predator Response}\right) = \log\left(\frac{P(\text{off})}{1-P(\text{off})}\right)_{\textit{Predator}} - \log\left(\frac{P(\text{off})}{1-P(\text{off})}\right)_{\textit{Mock}} \tag{3}$$

which can straightforwardly be computed for a particular experimental condition or genotype from logistic models by linear combination of the coefficients in β (*Equations 1–2*, with associated standard error and confidence intervals used for inference). In plain language this represents the change in odds of off lawn egg laying observed in the predator condition relative to the mock control. As natural logarithms are cumbersome for easy interpretation on plots, *Figures 5–9* use base 2 logarithms where each unit change corresponds to a twofold change in the predator response as defined in *Equation 3* above.

All linear hypotheses were computed using the glht function in the multcomp package in R with associated correction for multiple testing performed using the multivariate normal distribution (Z tests with the 'single step' method for generalized linear models, according to the simultaneous p-value estimation method of *Hothorn et al., 2008*). All statistical inference for differences between groups is performed on the logit scale but linear scale p(off) values are shown on plots for ease of interpretation.

## WormWatcher positional tracking data

Distance from body to center of arena over 20 hr of observation in WormWatcher assays was subjected to non-parametric bootstrap resampling with replacement for $10^5$ iterations with empirical 95% intervals determined using the quantile function in R. Significant changes to position with respect to time between conditions were inferred at $p<0.05$ where empirical bootstrapped intervals failed to overlap.

## Egg count data and P*nlp-29*::GFP fluorescence data

Average number of eggs per individual *C. elegans* in assays as well as logarithmic scale normalized fluorescence in *Figure 1* and *Figure 1—figure supplement 1* were tested for main effects and interactions between independent variables using general linear models using the lm function and the ANOVA function from the car package. To alleviate non-normality assessed by QQ Plot (qqnorm function in R) and heteroscedasticity in linear scale fluorescence data (assessed by Levene's tests, function leveneTest in R), data were $\log_2$ transformed and logarithmic scale data was subjected to further analysis. Omnibus effects in log-transformed data were determined by ANOVA. Where significant main effects or interactions were detected, post hoc linear hypothesis tests for differences between conditions were determined using the glht function in the multcomp package in R with associated correction for multiple testing performed using the multivariate t distribution (the 'single step' method for ANOVA/linear models according to the simultaneous p-value estimation method of *Hothorn et al., 2008*).

## Mouthform analysis

Changes to abundance of St or Eu *Pristionchus* was determined by Fisher's exact test.

## CLUSTAL alignment of DOP receptors

Alignment of receptors shown in *Figure 8* was performed using Clustal Omega on the EMBL-EBI server at https://www.ebi.ac.uk/Tools/msa/clustalo/ (*Sievers et al., 2011*).

## Acknowledgements

We would like to thank Nadia Haghani and Adeline Sov for their help in collecting the *Pristionchus* mouthform data. Anthony Fouad wrote custom code to help run the WormWatcher imaging setup as well as analyze the resulting images. We would also like to thank Kathleen Quach, Kirthi Reddy, Jess Haley, Wen Mai Wong, and Callum Walsh for contributing comments on this manuscript. This work was funded by a Graduate Research Fellowship from the National Science Foundation (AP), an Innovation grant from Kavli Institute of Brain and Mind (AP), and an NIH R01 MH113905 (SHC).

## Additional information

### Funding

| Funder | Grant reference number | Author |
|---|---|---|
| National Science Foundation | GRFP | Amy Pribadi |
| National Institute of Mental Health | MH113905 | Sreekanth H Chalasani |

The funders had no role in study design, data collection and interpretation, or the decision to submit the work for publication.

### Author contributions

Amy Pribadi, Conceptualization, Data curation, Formal analysis, Validation, Investigation, Methodology, Writing - original draft, Writing - review and editing; Michael A Rieger, Data curation, Formal analysis; Kaila Rosales, Data curation; Kirthi C Reddy, Resources, Data curation; Sreekanth H Chalasani, Conceptualization, Supervision, Funding acquisition, Writing - review and editing

### Author ORCIDs

Michael A Rieger http://orcid.org/0000-0003-4020-5476
Sreekanth H Chalasani http://orcid.org/0000-0003-2522-8338

### Ethics

This study was performed in strict accordance with the recommendation in the Guide for the Care and Use of Laboratory Animals of the National Institutes of Health.

### Decision letter and Author response

Decision letter https://doi.org/10.7554/eLife.83957.sa1
Author response https://doi.org/10.7554/eLife.83957.sa2

## Additional files

### Supplementary files
• MDAR checklist

### Data availability

All data generated or analysed during this study are included in the manuscript. Raw data for experiments in *Figures 1–9* and figure supplements are provided as source data files in MS Excel format. Analysis code for computation of associated effects can be found on the Shrek Lab GitHub: https://github.com/shreklab/PribadiEtAl2023 (copy archived at *Shrek Lab, 2023*).

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
