## [Editor Report]

In this important paper, Pribaldi and colleagues provide convincing evidence that locomotor and egg-laying behaviors of the nematode *C. elegans* can be altered by predation. They provide solid evidence that the neuromodulator dopamine is important for predator-evoked behavior, though further work will be necessary to understand how predator exposure might alter dopamine signaling. Because of the novelty of the behavioral findings and some important mechanistic insight, this work significantly advances the understanding of *C. elegans* neuroethology.

---

## [Decision Letter]

**Decision letter after peer review:**

Thank you for submitting your article "Dopamine signaling regulates predator-driven behavioral changes in *Caenorhabditis elegans*" for consideration by *eLife*. Your article has been reviewed by 2 peer reviewers, one of whom is a member of our Board of Reviewing Editors, and the evaluation has been overseen by Piali Sengupta as the Senior Editor. The reviewers have opted to remain anonymous.

Essential revisions:

1. Reviewer 2 raises some important concerns regarding the analysis and interpretation of your data. In a revised manuscript, please consider the null hypothesis that worms are simply dispersing and laying more eggs during the roaming state as a result of dopamine release. This will likely require some additional data analysis.

2. Please address reviewer 1's concern about the interpretation of the CEP-specific cat-2 rescue experiment.

Addressing these concerns does not require any new experiments.

*Reviewer #1 (Recommendations for the authors):*

I find the behavioral responses identified by the authors to be robust and compelling; the novelty of the finding is for me a large part of the appeal of the paper. Some aspects of the mechanistic dissection are not as strong as they could be (see detailed comments below) and I am surprised that the role of 5HT was not more carefully examined. Below are some specific comments that the authors may wish to consider, listed roughly in order of importance.

– It's interesting but a bit surprising that the eud-1 (St only) P. pacificus fail to elicit a behavioral response in *C. elegans*, even though they presumably still produce compounds that *C. elegans* perceives as aversive. I imagine the explanation is that this response isn't strong enough to bring about a change in egg distribution, but is it possible that the St form doesn't produce aversive chemical cues? If so, this would undermine the idea that the bite itself provides the trigger for behavioral change.

– The conclusions from the cat-2 rescue experiments may be overstated. Could the apparent importance of CEPs simply be a consequence of stronger expression of the CEP-specific promoter compared to the ADE/PDE promoter? It could be that the site of DA release is less important than the amount of release. The interpretation of this result should be toned down (lines 256-58, abstract, discussion).

– Figure 6 – I'm not sure I agree with the authors' interpretation of these results. The phenotype of the dop-1; dop-3 double is more than "slight", and the evidence for a meaningful role for dop-2 is weak.

– The exogenous DA experiments are fascinating but somewhat difficult to interpret. Do the authors think that exogenous DA is bypassing the effects of endogenous DA release (which would support the idea that DA signaling, at or downstream of its receptors, is altered in predator-exposed animals)? Or is it possible instead that exogenous DA is rapidly taken up by DA neurons and then released in a regulated way? In this case it's less clear how (or even if) DA signaling is being actively regulated.

– It's quite interesting that 5HT signaling is important for predator-induced behavior changes. The rationale for not studying this – that 5HT is already known to be important for egg-laying and in predator/prey behavior – is a little underwhelming. It doesn't seem obvious how the known functions of 5HT in worm behavior predict a straightforward explanation of its role here.

– It might be interesting to know whether nlp-29 has a role in the detection of the predator stimulus. The authors show that the behavioral change triggered by P. uniformus occurs without apparent induction of nlp-29::gfp, but it could be that this induction is weak/undetectable. It could also be that injury triggers an increase in nlp-29 release by the hypodermis without a strong upregulation of nlp-29 transcription.

– Line 243 – please rephrase (mutants don't contribute to the change, neurotransmitters do).

– Line 249 – please note that this expression pattern is specific to adult hermaphrodites.

– Lines 282-284 – this is an inappropriate overinterpretation of negative data. A more likely explanation for the lack of statistical support for early changes in behavior is that the experiment is underpowered.

– Lines 285-6 – "locomotor response" suggests changes in specific features of motor behavior – "navigational response" might be better here.

– Line 362 – Please provide additional details on the methods here. To conclude that the effect "lasts at least 6 hours" it would be necessary to examine the location of animals and/or eggs laid during a short window around the 6 hr time point, not cumulatively from 0-6 hours. It's hard to tell which was the case here.

– Lines 443-5 – word(s) missing from this sentence?

– Line 476 – this sentence needs editing.

– Figure S1 – legends for (b) and (c) are switched.

– Figure 6-S4 is mislabled as Figure 4-S4.

Reviewer #2 (Recommendations for the authors):

I think this is a great experimental paradigm, with a nice progression of genetic experiments to identify the role of dopamine release from CEP neurons as causing the altered behavior in response to predation. However, I think how the data are quantified and interpreted should be altered, as mentioned in my prior comments. I realize the time resolution of the experiments are not sufficient to quantify random walk dynamics, and I am normally loathed to request more experiments, especially since so many have been performed here, but perhaps the data could be reanalyzed in a way that more accurately reflects the underlying phenomenon.

For example, the authors claim more eggs are laid off the lawn, and quantify this in Figure 1-S1. Why did you not do this for all the data? This is simple, and could be easily quantified with a simple chi-square analysis. The next question is, why? Since worms inherently perform random walks, the null hypothesis would be that their effective diffusion constant has changed. The drive to perform random walks is fundamental strategy of worm navigation, even while moving toward favorable conditions. A nice quantification of this property can be found in Klein et al., *eLife*, 2017.

This is especially relevant since we know that the worm's effective diffusion constant is different on and off food, and that neuromodulation alters these statistics. I realize your movies don't have sufficient resolution to capture these dynamics, but you do have the spatial distribution of eggs, correct? Why not plot that distribution for the different conditions, rather than quantifying the relative distance from the food edge, which is nebulous given the effect size? You could then compare distributions, and do a bootstrap analysis to verify whether changes in distribution are different? Given what's already known about dopamine and egg laying (Cermak et al., 2020), as well as decades of work quantifying the worm's random walk behavior, a shift in the statistics of this underlying stochastic motor program should be considered the null hypothesis.

I appreciate the Appendix with the statistics, but in the absence of the raw data, it's really impossible to validate if any of this is accurate. Surely these analyses were based upon the analysis of text files/tables of data, correct? Why not include them?

[Editors' note: further revisions were suggested prior to acceptance, as described below.]

Thank you for resubmitting your work entitled "Dopamine signaling regulates predator-driven changes in *Caenorhabditis elegans'* egg laying behavior" for further consideration by *eLife*. Your revised article has been evaluated by Piali Sengupta (Senior Editor) and a Reviewing Editor.

Both reviewers feel that the revised manuscript has been substantially improved. However, before publication, we would like you to consider Reviewer #1's point about the role of dopamine in bringing about predator-induced changes in behavior. The bulk of your evidence points to a function for DA in egg-laying per se (which of course is already well-established), with its role in changes in behavior upon predator exposure being much more subtle. We suggest being a bit more cautious in some of your interpretations.

Each reviewer also has several minor comments/suggestions that you may wish to address.

*Reviewer #1 (Recommendations for the authors):*

The revised manuscript is significantly improved, largely due to the change in data analysis suggested by reviewer #2. The demonstration of the persistent effects of predator exposure on locomotor and egg-laying behavior is robust and fascinating.

My biggest concern about the revised manuscript is that the role of dopamine in predator-induced behavioral change is a bit overstated. Nearly all of the experiments in the paper now clearly show that the primary role of dopamine in this paradigm is to modulate baseline P(off). The direct evidence that DA is required for predator-induced changes is relatively slim. To my mind, the only compelling result is the dop-1;2;3 triple mutant in Figure 9c. In general, the experiments in Figures8 and 9 are tricky to interpret because the baseline P(off) is quite a bit lower here than in previous experiments. This raises the concern that floor effects are limiting the ability to see further reductions in baseline P(off) in mutants, which could in turn lead to artificially low values for fold-change in P(off) in the predator-exposed condition.

---

## [Author Response]

Essential revisions:1. Reviewer 2 raises some important concerns regarding the analysis and interpretation of your data. In a revised manuscript, please consider the null hypothesis that worms are simply dispersing and laying more eggs during the roaming state as a result of dopamine release. This will likely require some additional data analysis.

We have fully re-analyzed egg assay data using the quantification of # of eggs laid off and on lawns as the primary metric for discerning effects across conditions (see extensive treatment in response to Reviewer 2). This is both a powerful and easily digestible statistical paradigm as we are now primarily concerned with the probability of eggs being laid on and off lawn and the interactions of various conditions on this probability.

As a result of this, a number of concerns of the reviewers specifically regarding dopamine synthesis and mutants in dopaminergic signaling are well founded. We find on the whole that mutants in dopamine synthesis do not in fact show reproducible blunting of the predator response itself, but instead seem to blunt the overall probability of laying eggs off the lawn even in control conditions. However, we have found significant effects on predator exposure with respect to combinations of dopamine receptors mutants. In light of this re-analysis, we have re-ordered the paper, replotted figures, and in general overhauled the way we treat these data. Again, please see the extensive comment to Reviewer #2.

2. Please address reviewer 1's concern about the interpretation of the CEP-specific cat-2 rescue experiment.

In light of the statistical overhaul noted above, we are not highlighting the role in these neurons in altering the predator effect itself. It is clear that *cat-2* mutants blunt the baseline probability of off-lawn laying. When looking at proportions of eggs laid off the lawn, rescue in ADE/PDE partially restores this background, and CEP further elevates the probability of off-lawn laying in the control condition. All show an increased proportion of eggs laid off lawn when predator is present. Please see lines 388 – 404 for the description and discussion of these data.

Addressing these concerns does not require any new experiments.Reviewer #1 (Recommendations for the authors):I find the behavioral responses identified by the authors to be robust and compelling; the novelty of the finding is for me a large part of the appeal of the paper. Some aspects of the mechanistic dissection are not as strong as they could be (see detailed comments below) and I am surprised that the role of 5HT was not more carefully examined. Below are some specific comments that the authors may wish to consider, listed roughly in order of importance.– It's interesting but a bit surprising that the eud-1 (St only) *P. pacificus* fail to elicit a behavioral response in *C. elegans*, even though they presumably still produce compounds that *C. elegans* perceives as aversive. I imagine the explanation is that this response isn't strong enough to bring about a change in egg distribution, but is it possible that the St form doesn't produce aversive chemical cues? If so, this would undermine the idea that the bite itself provides the trigger for behavioral change.

We find that conditioning a lawn with *P. uniformis* secretions is not sufficient to drive a change the probability of *C. elegans* off-lawn egg laying (Figure 1—figure supplement 6). We do agree with the reviewer that eud-1 mutants might still produce aversive compounds for *C. elegans* prey. However, our egg-laying assay is unable to identify the behavioral changes. We have included a statement about eud-1 mutants in our revised manuscript Line 177-179.

New manuscript Line 177: “These data indicate that interactions between eud-1 mutants and prey (secretions, contacts, and others) are unable to alter the locations of *C. elegans* eggs.”

– The conclusions from the cat-2 rescue experiments may be overstated. Could the apparent importance of CEPs simply be a consequence of stronger expression of the CEP-specific promoter compared to the ADE/PDE promoter? It could be that the site of DA release is less important than the amount of release. The interpretation of this result should be toned down (lines 256-58, abstract, discussion).

As this paper has gone through restructuring upon re-analysis of egg data for clarity we will copy original text into this rebuttal in order to reference old and new line numbers:

Original manuscript "We found that both transgenic animals restored responses to P. uniformis males, but CEP-specific expression was able to increase off-lawn egg laying to near wildtype levels (Figure 4c). These results indicate dopamine synthesis from CEPs likely play a major role in altering egg location upon exposure to P. uniformis males."

Upon re-analysis we find that primarily the effect of *cat-2* mutant allele(s) is to dampen the baseline of off-lawn laying: i.e., reductions to the probability that *C. elegans* will lay eggs off the lawn in control (non-predator-exposed) conditions. We find (Figure 6g-i) that rescue of CAT-2 in ADE/PDE partially restores off-lawn laying under control conditions and rescue in CEP restores this behavior to beyond that of wild-type animals. This, we believe, likely underlies the apparent restoration of predator-evoked levels of off-lawn laying, or at least the two phenotypes (changes to baseline off-lawn behavior, changes to predator-evoked response) are challenging to disentangle.

We thank the reviewer for this observation. Yes, it is case that differences in expression levels brought about by using different promoters could be driving differences in rescue between the two cell populations. Due to the findings upon re-analysis and the reviewer's observation, we have softened this tone in general.

New manuscript 397 – 404: "The cohorts of cat-2 mutants used in Figure 6a-b, Figure 6c-d, as well as the results shown in Figure 7 described below, indicate that changes to the underlying probability of laying eggs off the lawn is likely driving any observed effects to predator response. Additionally, differences in promoter strength used to drive expression of cat-2 may explain why dopaminergic cell types show differing ability to restore baseline P(off). Nevertheless, it is clear that re-expression of CAT-2 protein in either ADE/PDE or CEP only is sufficient to at least partially restore baseline off lawn egg laying behavior."

– Figure 6 – I'm not sure I agree with the authors' interpretation of these results. The phenotype of the dop-1; dop-3 double is more than "slight", and the evidence for a meaningful role for dop-2 is weak.

Original manuscript Figure 6 -> New manuscript Figures 8 and 9.

These data have been fully re-analyzed using a binomial model to estimate the probability of off-lawn laying (see response to Reviewer #2). As a result, we show in new main figures 8 and 9 the single and double/triple mutant data.

When analyzed in terms of probability of off-lawn laying, there are clearly effects to the background level of off-lawn laying in control conditions. To some degree, the situation is clearest in the triple vs. quadruple mutant (new Figure 9d-f). In the triple mutant, the background (control) level of off-lawn laying is comparable to WT, but the predator response is blunted, whereas in the quadruple mutant, the control level is also depressed. Thus, although the predator response is lower in an absolute sense, it is not lower in fold change relative to its own control. We acknowledge these are **not** paired data, however, we believe off-lawn laying in the control condition is still a reasonable estimate of these mutants’ “baseline” of activity and a logistic regression/binomial generalized linear models can still estimate the confidence intervals of the magnitude of fold change between control and predator conditions.

With respect to the *dop-1;dop-3* double mutant, we agree, this is not “slight”. In light of re-analysis, again, these results have been re-interpreted. The double mutant’s fold change in Predator – Control is a useful way to rank them with respect to their effect on predator response. This is shown in new Figure 9c. *dop-3;dop-4* has the most comparable fold change compared to WT, whereas *dop-1;dop-3* and *dop-2;dop-3* sit at the lowest magnitude fold change. We hope this presentation makes these complex effects clearer to the reader.

Our re-interpreted results are presented in the new manuscript as follows:

New manuscript (lines 465-474): “ These combinations also had differing effects on baseline P(off) in Mock controls. Dop-1;dop-4 mutants were the most similar to WT. dop-1;dop-2, dop-2;dop-4 and dop-2;dop-3 all showed elevation of baseline off lawn egg laying activity relative to WT, and dop-1;dop-3 and dop-3;dop-4 showed reductions to baseline P(off). The agnitude of Predator Response in these mutant combinations is shown ordered from highest to lowest in Figure 9b. WT and dop-3;dop-4 double mutants show the highest fold change increase in P(off) relative to their respective Mock controls. All combinations containing dop-4 rank intermediate with dop-2;dop-3 and dop-1;dop-3 ranking lowest. Other than dop-3;dop-4, all other combinations showed reduction to predator response relative to WT. “

– The exogenous DA experiments are fascinating but somewhat difficult to interpret. Do the authors think that exogenous DA is bypassing the effects of endogenous DA release (which would support the idea that DA signaling, at or downstream of its receptors, is altered in predator-exposed animals)? Or is it possible instead that exogenous DA is rapidly taken up by DA neurons and then released in a regulated way? In this case it’s less clear how (or even if) DA signaling is being actively regulated.

Old manuscript Figure 8 -> New manuscript Figure 7

In light of re-analysis, it seems clear that what DA is rescuing is primarily the baseline of off-lawn laying activity under control conditions. In other words, *cat-2* mutant rarely lay eggs off lawn at all. Thus, they increase the proportion of off-lawn eggs after predator exposure, but this is at lower proportion than WT given their baseline low activity of laying eggs off the lawn. When exogenous DA is supplemented, the baseline in the control condition comes back up to WT levels and even exceeds it.

We are unsure about the mechanism of this effect and would require additional experiments. One possible experiment would be to cross the *cat-2* and *dat-1* mutant backgrounds. If DA is still able to restore normal off-lawn egg laying in this double mutant background, it is then likely because exogenous dopamine is bypassing the re-uptake and regulated release. For the scope of this paper, however, we would argue that the point of this experiment would be say that it is *cat-2* ‘s role as an upstream enzyme in dopamine synthesis that underlies changes to off-lawn laying and not some other effect of mutation of this gene. We have speculated a bit about possible reasons for this, as well as the roles dopaminergic receptors may be playing, in our Discussion (New manuscript Lines 554-589).

– It’s quite interesting that 5HT signaling is important for predator-induced behavior changes. The rationale for not studying this – that 5HT is already known to be important for egg-laying and in predator/prey behavior – is a little underwhelming. It doesn’t seem obvious how the known functions of 5HT in worm behavior predict a straightforward explanation of its role here.

We wanted to focus this manuscript on only one neurotransmitter rather than attempting to address both; we believe that 5HT signaling in this context is worth pursuing in the future but believe it is out of the scope for this current manuscript. Clarified this point, see line 377-379 new manuscript:

New manuscript line 377: “We focused our remaining studies on dopaminergic signaling, but future work will investigate the role of serotonin signaling as serotonin has been previously shown to modify egg laying behavior [50,51].”

– It might be interesting to know whether nlp-29 has a role in the detection of the predator stimulus. The authors show that the behavioral change triggered by P. uniformus occurs without apparent induction of nlp-29::gfp, but it could be that this induction is weak/undetectable. It could also be that injury triggers an increase in nlp-29 release by the hypodermis without a strong upregulation of nlp-29 transcription.

It is possible that nlp-29::GFP induction by *P. uniformis* is present but at undetectable levels. We also believe that not all injuries to the cuticle can induce nlp-29::GFP; it is possible that only injuries that pierce deeply into the hypodermis are capable of inducing nlp-29::GFP and that shallower injuries do not. However, we believe that, *P. uniformis* damages the cuticle less than *P. pacificus*, so grievous bodily harm can be ruled out as a main driver of behavioral change.

– Line 243 – please rephrase (mutants don’t contribute to the change, neurotransmitters do).

Original manuscript: " These data show that mutants in both serotonin and dopamine synthesis contribute to the egg location change in response to P. uniformis males."

New manuscript line 371: "Taken together, these data show loss of biogenic amine neurotransmitters can modify off lawn egg laying behavior, attenuating or even increasing the observed response to predator, though these two phenomena were not so clearly separable. Loss of both dopamine and serotonin neurotransmitters in cat-1 mutants, however, not only reduced the general probability of off lawn laying but also contributed to the largest blunting of the predator response.”

– Line 249 – please note that this expression pattern is specific to adult hermaphrodites.

Original manuscript: "In *C. elegans*, CAT-2 is expressed by eight neurons (four CEPs, two ADEs and PDEs), and dopamine signaling has been previously shown to affect modulation of locomotion as well as learning [50]."

New manuscript line 383: "In *C. elegans* adult hermaphrodites, CAT-2 protein is expressed by eight neurons (four CEPs, two ADEs and PDEs), and dopamine signaling has been previously shown to affect modulation of locomotion as well as learning [43]. "

– Lines 282-284 – this is an inappropriate overinterpretation of negative data. A more likely explanation for the lack of statistical support for early changes in behavior is that the experiment is underpowered.

Original manuscript: "This could be due to incomplete knockdown of dopamine synthesis in cat-2(e1112) [52], or due to an additional role for dopamine in altering egg laying dynamics [53] during predator exposure."

We have removed this interpretation. WormWatcher data has been re-analyzed using a non-parametric bootstrap approach which makes this interpretation unsound in any case. See Figure 6—figure supplement 1.

– Lines 285-6 – "locomotor response" suggests changes in specific features of motor behavior – "navigational response" might be better here.

Original manuscript: "As reducing the amount of available dopamine in the *cat-2* mutant depressed locomotor response to predator, we hypothesized that increasing dopamine would increase the response to predator."

New manuscript: This general and following interpretations of what occurs with *dat-1* mutation are no longer sound given re-analysis of the WormWatcher data using non-parametric bootstrap approach, and we have removed this particular transitional sentence from the new manuscript.

– Line 362 – Please provide additional details on the methods here. To conclude that the effect "lasts at least 6 hours" it would be necessary to examine the location of animals and/or eggs laid during a short window around the 6 hr time point, not cumulatively from 0-6 hours. It's hard to tell which was the case here.

We have fully shown the data in this figure over the course of 6 hours while addressing the nature of the temporal aspect of these phenotypes in light of re-analysis as proportion of eggs off lawn. See treatment in new manuscript lines: 289 – 316. There is an interesting interaction with time in these data that is worth nothing.

For orientation, this y-axis is the probability of off lawn egg laying. Trend lines over time are shown in four panels because plotting all these data in a single panel would be difficult to interpret and little messy. Data points however are only shown once per plot as this is a single experiment so as to not give the impression that data points are used for multiple analyses more than once.

Fitting the three-way interaction of time, predator exposure, and lawn condition, shows interesting effects. Namely: artificial streaking shows a positive relationship with laying eggs off lawn over time: worms are venture out to lay eggs more often over time. After predator exposure, this slope is flatter and worms are simply laying eggs off the lawn across all time points. When these two conditions are combined, not only is the time evolution flatter but also a much larger proportion of eggs are laid off lawn.

We feel the current treatment provides the best understanding of the data and is a natural way to understand competing interests for the worms: recent predator experience which may promote laying eggs at a distance, as well as bacterial concentrations existing off the main lawn which may have drawn prey to those locations. Finally, it is clearer that with predator exposure, the effect is already true even at 1 hour and persists relatively evenly across the 6 hours.

– Lines 443-5 – word(s) missing from this sentence?

Sections about mammalian homology as well as the actions of the triple and quadruple mutant are now split up. See new manuscript:

New manuscript Line 445: "*C. elegans* DOP-1 is a homolog of the mammalian D1-like receptors and DOP-2/3 are homologs of mammalian D2-like receptors [43]. DOP-4 is also D1-like, however this receptor belongs to a unique invertebrate family of D1-like including receptors found in *Drosophila melanogaster* and *Apis mellifera* [57]. "

New manuscript Line 477: "To test the hypothesis of the presence or absence of just dop-4 influencing predator-evoked behavior, we performed an experiment comparing triple mutant animals in dop-1;dop-2;dop-3 to quadruple mutants of all four receptors (Figure 9c). Once again, the quadruple mutant showed reduction to the baseline P(off) in the Mock control as in Figure 8, however, the triple mutant showed a comparable level of off lawn laying in the mock condition relative to WT.”

– Line 476 – this sentence needs editing.

The Ideas and Speculation section of our Discussion has been restructured.

– Figure S1 – legends for (b) and (c) are switched.

All figures and legends have been overhauled and current legend for Figure 1—figure supplement 1-4 should have all panels correctly identified.

– Figure 6-S4 is mislabled as Figure 4-S4.

Single dopamine receptors have been moved to new Figure 8, and all figures have been renamed to follow correct *eLife* conventions.

Reviewer #2 (Recommendations for the authors):I think this is a great experimental paradigm, with a nice progression of genetic experiments to identify the role of dopamine release from CEP neurons as causing the altered behavior in response to predation. However, I think how the data are quantified and interpreted should be altered, as mentioned in my prior comments. I realize the time resolution of the experiments are not sufficient to quantify random walk dynamics, and I am normally loathed to request more experiments, especially since so many have been performed here, but perhaps the data could be reanalyzed in a way that more accurately reflects the underlying phenomenon.For example, the authors claim more eggs are laid off the lawn, and quantify this in Figure 1-S1. Why did you not do this for all the data? This is simple, and could be easily quantified with a simple chi-square analysis. The next question is, why? Since worms inherently perform random walks, the null hypothesis would be that their effective diffusion constant has changed. The drive to perform random walks is fundamental strategy of worm navigation, even while moving toward favorable conditions. A nice quantification of this property can be found in Klein et al., eLife, 2017.This is especially relevant since we know that the worm's effective diffusion constant is different on and off food, and that neuromodulation alters these statistics. I realize your movies don't have sufficient resolution to capture these dynamics, but you do have the spatial distribution of eggs, correct? Why not plot that distribution for the different conditions, rather than quantifying the relative distance from the food edge, which is nebulous given the effect size? You could then compare distributions, and do a bootstrap analysis to verify whether changes in distribution are different? Given what's already known about dopamine and egg laying (Cermak et al., 2020), as well as decades of work quantifying the worm's random walk behavior, a shift in the statistics of this underlying stochastic motor program should be considered the null hypothesis.

Thank you to the reviewer for these excellent points. We have overhauled the analysis performed on these data which highlight some of the concerns that the reviewer has here.

So, it is clear that (a) eggs have a distribution, and (b) these distributional properties are oversimplified by collapsing to a simple metric such as the mean/median/central tendency of eggs away from the lawn. A bootstrap analysis of these distributions is actually not the greatest technique here. Bootstrapping still requires a bootstrap statistic. If, for example, that statistic were the mean or the median, then the bootstrap derived empirical intervals for that statistic would still reflect estimation of that quantity and fail to capture other interesting features of the data such as the changes to dispersal of the eggs.

As it is known that worms detect the lawn edge, the binary off/on-lawn is a very reasonable approach for quantifying the eggs. When we performed this analysis, we find that indeed *cat-2* mutants do respond to predator by laying eggs more frequently off the lawn (see new Figure 6). Consider the data shown in new Figure 6C for example, which is an experiment looking at two *cat-2* mutant alleles. If we analyze the mean, the coefficient of variation, as well as the upper and lower quartiles of the distribution, all report that the *cat-2* mutants are “not” altering their distributions with respect to exposure to predator. However, if you simply tabulate the quantity of eggs laid off the lawn, you see quite clearly that they do the respond. The reason for this statistical phenomenon is that the *cat-2* mutants very rarely lay eggs off the lawn at all, even under control conditions, whereas WT animals venture off the lawn to lay eggs at least some of the time. So, when *cat-2* mutants are exposed to predator they do in fact increase the proportion of their eggs laid off the lawn relative to this low background rate, and due to the rarity of these events, this effect is most simply and easily captured by quantifying the number of eggs laid on and off the lawn. This might be summarized by a distributional parameter such as a quantile, but this would require decision making about which region of the distribution to look in. This is possible but seems overly complicated or biased in light of the simpler metric of the lawn boundary and the numbers of eggs on and off the lawn.

Thus, we have proceeded in our egg assays to analysis the [# off lawn eggs, # on lawn eggs]. For a simple experiment with few independent variables, we agree that a Fisher’s exact or chi-square approach is reasonable. However, we have complex effects some of which include two- and three-way interactions between variables. Instead, we have turned to using logistic regression/binomial generalized linear modeling which gives us analogous tools to ANOVA in the form of likelihood ratio statistics/analysis of deviance. Binomial modeling uses the statistic of the Odds Ratio which is also a very natural metric corresponding to the “chance of success” – in our case defined as the probability of leaving the lawn to lay eggs versus staying on the lawn to lay eggs. This paradigm give us a simple and powerful way to quantify, and analyze effects in the data. In the binomial model, the input N are the eggs themselves which have a value of 1 or 0 depending on whether they are “success” (defined here as off the lawn) or “failure” (defined here as on the lawn), which are stratified by experimental replicate. Thus, this approach takes into account every egg in every arena and thus greatly boosts statistical power. These changes are reflected in reported statistics throughout the manuscript and in the figure legends/methods. We are now enabled to perform inference about changes to : the background level of off-lawn laying in control conditions, which varies, and “double Δ” interaction effects: i.e., change in the magnitude of change between predator and control odds of off-lawn laying, which reflects what we are terming a Predator Response (see new Methods/Statistics section, especially Equation #3). Effectively this is the change in the numbers of eggs laid off lawn in predator-exposed conditions controlling for the Mock control level of off lawn laying.

I appreciate the Appendix with the statistics, but in the absence of the raw data, it's really impossible to validate if any of this is accurate. Surely these analyses were based upon the analysis of text files/tables of data, correct? Why not include them?

As our statistical analyses have been overhauled, all raw data files are now included as source data files according to *eLife* submission standards.

[Editors' note: further revisions were suggested prior to acceptance, as described below.]

Both reviewers feel that the revised manuscript has been substantially improved. However, before publication, we would like you to consider Reviewer #1's point about the role of dopamine in bringing about predator-induced changes in behavior. The bulk of your evidence points to a function for DA in egg-laying per se (which of course is already well-established), with its role in changes in behavior upon predator exposure being much more subtle. We suggest being a bit more cautious in some of your interpretations.Each reviewer also has several minor comments/suggestions that you may wish to address.Reviewer #1 (Recommendations for the authors):The revised manuscript is significantly improved, largely due to the change in data analysis suggested by reviewer #2. The demonstration of the persistent effects of predator exposure on locomotor and egg-laying behavior is robust and fascinating.

We thank the reviewer for their feedback.

My biggest concern about the revised manuscript is that the role of dopamine in predator-induced behavioral change is a bit overstated. Nearly all of the experiments in the paper now clearly show that the primary role of dopamine in this paradigm is to modulate baseline P(off). The direct evidence that DA is required for predator-induced changes is relatively slim. To my mind, the only compelling result is the dop-1;2;3 triple mutant in Figure 9c. In general, the experiments in Figures8 and 9 are tricky to interpret because the baseline P(off) is quite a bit lower here than in previous experiments. This raises the concern that floor effects are limiting the ability to see further reductions in baseline P(off) in mutants, which could in turn lead to artificially low values for fold-change in P(off) in the predator-exposed condition.

The background, as assessed from the “absolute” control i.e. a WT animal exposed only to other *C. elegans* does vary a bit from assay setup to assay setup. In Figure 1, N2 animals are not used, GFP fluorescent animals are used to distinguish *C. elegans* eggs from the eggs of other predators. When we switch to using *P. uniformis* males throughout the remaining studies in this work, we were able to switch to N2 as the reference strain. However, differences in background between Figure 1 and others may be due to this strain difference.

In the remainder of the studies with the exception of Figure 4 and 5 and 7 where animals are tested for eggs only after exposure to predator (learning assays), in which the background may be affected by this change in setup, the reviewer will note that the background in Figure 8 and 9 seems not too far off the range of variability observed in other experiments. Explicitly:

WT/mock, Figure 3: P(off) estimate at 0.09

Figure 6A: P(off) estimate at 0.21

Figure 6C: P(off) estimate at 0.15

Figure 6E: P(off) estimate at 0.34

Figure 8B: P(off) estimate at 0.05

Figure 9A: P(off) estimate at 0.15

Figure 9C: P(off) estimate at 0.10

In fact, given these estimates, it is the rescue assay in Figure 6E that seems to be surprisingly high.

We do not know what accounts for this variability. It may be that there are aspects of the animal’s internal state at assay start that contribute to it. But we do not think it is unreasonable to perform inference within an experiment with respect to its own background. However future exploration of the relationship of an animal’s internal state to its propensity for laying eggs off the lawn merits study.